# In vitro reconstitution of *Escherichia coli* divisome activation

Philipp Radler[1,4], Natalia Baranova[1,2,4], Paulo Caldas[3], Christoph Sommer [1], Mar López-Pelegrín[1], David Michalik [1] & Martin Loose [1✉]

The actin-homologue FtsA is essential for *E. coli* cell division, as it links FtsZ filaments in the Z-ring to transmembrane proteins. FtsA is thought to initiate cell constriction by switching from an inactive polymeric to an active monomeric conformation, which recruits downstream proteins and stabilizes the Z-ring. However, direct biochemical evidence for this mechanism is missing. Here, we use reconstitution experiments and quantitative fluorescence microscopy to study divisome activation in vitro. By comparing wild-type FtsA with FtsA R286W, we find that this hyperactive mutant outperforms FtsA WT in replicating FtsZ treadmilling dynamics, FtsZ filament stabilization and recruitment of FtsN. We could attribute these differences to a faster exchange and denser packing of FtsA R286W below FtsZ filaments. Using FRET microscopy, we also find that FtsN binding promotes FtsA self-interaction. We propose that in the active divisome FtsA and FtsN exist as a dynamic copolymer that follows treadmilling filaments of FtsZ.

[1] Institute for Science and Technology Austria (IST Austria), Klosterneuburg, Austria. [2] University of Vienna, Department of Pharmaceutical Sciences, Vienna, Austria. [3] UCIBIO—Applied Molecular Biosciences Unit, Department of Life Sciences, NOVA School of Science and Technology, Universidade Nova de Lisboa, Caparica, Portugal. [4] These authors contributed equally: Philipp Radler, Natalia Baranova. ✉email: martin.loose@ist.ac.at

Bacteria have intricate intracellular organizations, where different proteins localize to distinct sites in a tightly regulated, highly dynamic manner. The molecular mechanisms that give rise to these complex spatiotemporal dynamics are often unknown. This is in particular true for the divisome, a highly complex protein machinery that accomplishes cell division with remarkable precision[1]. The divisome consists of more than a dozen different proteins that assemble in a step-like manner. Divisome assembly in *E. coli* is initiated by the simultaneous accumulation of FtsZ, FtsA and ZipA at midcell, where they organize into the Z-ring, a composite cytoskeletal structure of treadmilling filaments at the inner face of the cytoplasmic membrane (Fig. 1a). In a second step, this dynamic Z-ring recruits cell division proteins to the division plane and promotes their homogeneous distribution around the circumference of the

cell[2]. Finally, the cell starts to constrict while generating two new cell poles splitting the dividing cell in two. Although the biochemical network underlying cell division is now well studied[3], how the membrane anchors of FtsZ control the timing of recruitment and activation of cell division proteins located in the cell membrane is currently unknown[4–6].

The actin-homologue FtsA is widely conserved and generally considered to be the more important membrane linker for FtsZ filaments[7,8]. It can reversibly bind to the membrane via a C-terminal amphipathic helix, where it recruits FtsZ filaments by binding to their C-terminal peptides (Fig. 1b). In return, FtsZ filaments also determine the spatiotemporal distribution of FtsA on the membrane[9,10]. FtsA was found to self-interact to oligomerize into actin-like single or double protofilaments[11] as well as membrane-bound minirings composed of 12 FtsA monomers

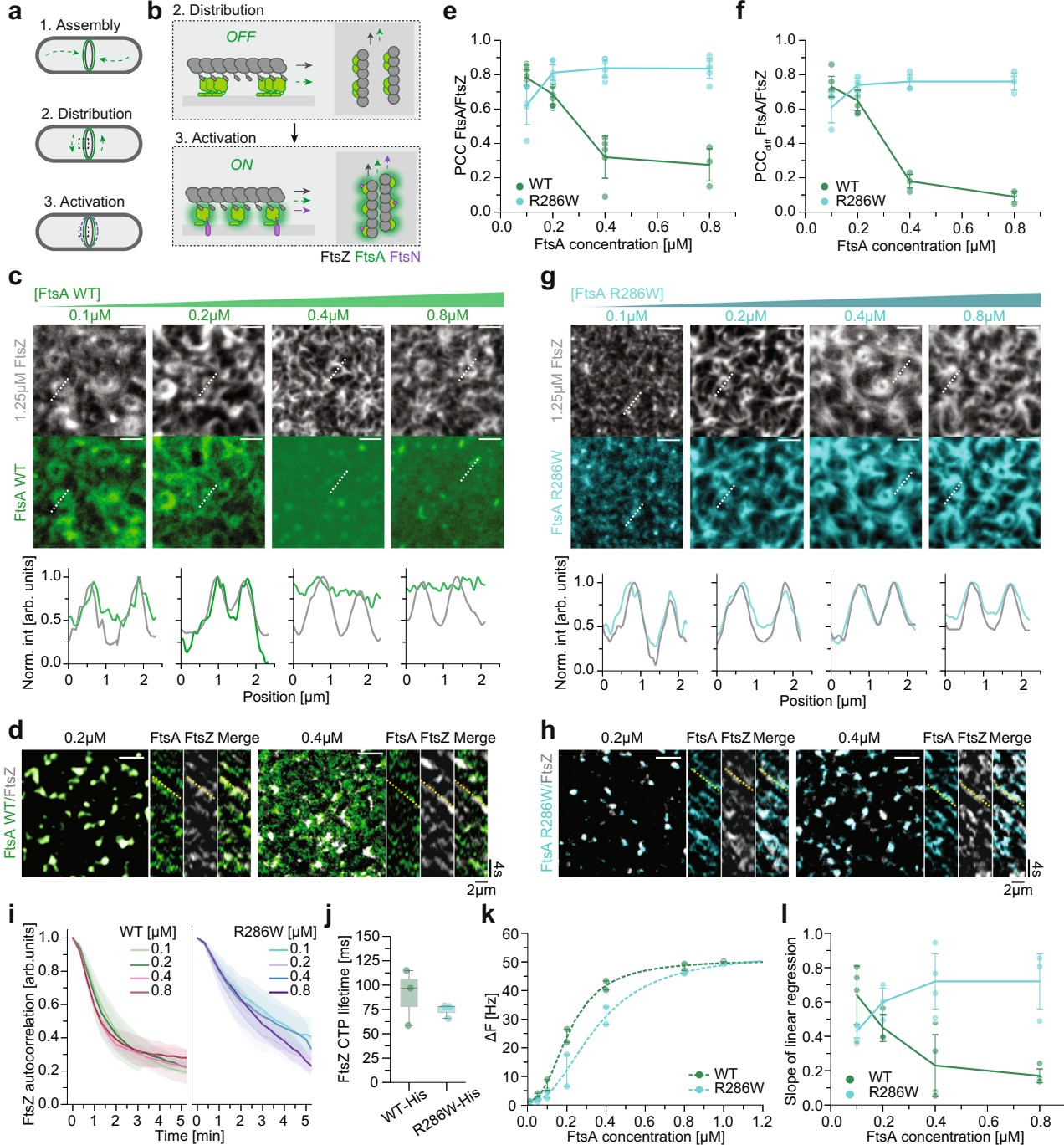

**Fig. 1 Membrane patterning of FtsA by treadmilling filaments of FtsZ. a, b** Current model of divisome activation. FtsA anchors FtsZ to the cytosolic membrane of *E. coli,* and treadmilling FtsZ filaments distribute FtsA around midcell. **b** Side (left) and top-view (right). FtsN binding is expected to switch FtsA from an oligomeric *off* to the monomeric *on* state. **c** Representative micrographs of Alexa488-FtsZ (grey) and Cy5-FtsA WT (green) at increasing FtsA and constant FtsZ concentrations. Intensity profiles correspond to dashed white lines. **d** Representative micrographs showing merged differential image of FtsZ with 0.2 μM (left) and 0.4 μM (right) FtsA. Yellow lines in kymographs indicate the slope for treadmilling FtsZ. **e** Colocalization of WT (green) and R286W (cyan) with FtsZ quantified by PCC. *n*(WT) = 11/9/5/3 and *n*(R286W) = 5/8/6/6 for 0.1/0.2/0.4/0.8 μM respectively. **f** Dynamic colocalization of WT (green) and R286W (cyan) with FtsZ quantified by PCC_diff. *n*(WT) = 6/6/3/3 and *n*(R286W) = 3/5/4/4. **g** Representative micrographs of Alexa488-FtsZ (grey) and Cy5-FtsA R286W (cyan) at increasing FtsA and constant FtsZ concentrations. Intensity profiles correspond to dashed white lines. **h** Representative merged differential images of FtsZ and 0.2 μM (left) or 0.4 μM (right) FtsA R286W. Yellow lines in kymographs indicate the slope for treadmilling FtsZ. **i** The FtsZ network is more persistent with FtsA R286W, showing a slower autocorrelation decay. *n*(WT) = 8/8/4/ 2 and *n*(R286W) = 7/7/6/6. **j** The FtsZ-C-terminal peptide has the same lifetime on 1% Tris-NTA lipids membranes with WT-His6 or R286W-His6. *n*(WT/R286W) = 3. **k** QCM-D experiments reveal that R286W binds slightly weaker to bilayers compared to WT. However, membranes are saturated at 0.8 μM for both FtsAs. *n*(WT/R286W) = 2. **l** The slope of the linear regression is proportional to the [FtsZ] vs [FtsA] ratio. *n*(WT) = 4/3/4/4 and *n*(R286W) = 3/3/4/3. Scale bars are 2 μm. Dots represent independent experiments, thick lines indicate means, error bars or shaded areas (**i**) depict the standard deviation. The boxes indicate the 25–75th percentiles, whiskers show the maximum/minimum values within the standard deviation and the midline indicates the median. Source data are provided as a Source Data file.

with a diameter of about 20 nm[12]. In addition, FtsA binds to many other proteins of the divisome, including FtsN, FtsQ, FtsX and FtsW[4,6,13–17] highlighting its indispensable role for cell division. While FtsA is essential in *E. coli*, several FtsA mutants, such as FtsA R286W, can compensate for the loss of other essential components of the divisome, including the alternative membrane-anchor ZipA, but also FtsEX, FtsN, FtsQ and FtsK[7,18,19]. These suppressor mutants were also found to facilitate the recruitment of division proteins and stabilize the Z-ring, leading to premature division[5,8,18–20]. Importantly, these mutations are located at or near the binding interface between two FtsA subunits and result in a reduced self-interaction in yeast two-hybrid assays as well as the absence of cytoplasmic rods when membrane-binding deficient proteins are overexpressed[19]. Together, these observations led to the idea that divisome maturation and cell constriction depends on the switch of FtsA from an inactive, polymeric state to the active, monomeric form[3,16,19] (Fig. 1b). As oligomerization and recruitment of downstream proteins appear to be mutually exclusive, binding of FtsN and other divisome proteins would depolymerize FtsA filaments and activate the cell division machinery, while FtsA suppressor mutants were proposed to resemble the monomeric, already active state of the protein.

While this model is consistent with many observations made in vivo, direct biochemical evidence for FtsA's different activity states, their molecular properties and the mechanism of their conversion remains missing so far. In addition, a quantitative characterization of the dynamic interplay between cell division proteins on the cytoplasmic membrane in vivo has so far been challenging due to limited availability of fully functional fluorescent fusion proteins. Here, we have reconstituted the interactions between treadmilling filaments of FtsZ, its membrane-anchor FtsA and the cytoplasmic peptide of the late division protein FtsN on membrane surfaces in vitro. By comparing the properties of wild-type FtsA and FtsA R286W, a gain-of-function mutant proposed to reside in a constant active state, we provide answers to two fundamental questions about the role of FtsA for divisome maturation and initiation of cell constriction: first, what is the relationship between the activity state of FtsA, its self-interaction and the recruitment of downstream proteins? And second, how does activation of FtsA affect the spatiotemporal organization of itself and that of FtsZ filaments on the membrane? By answering these questions, we shed light on the role of FtsA filaments for bacterial cell division and also identify general requirements we believe to be important for the propagation of biochemical signals in living cells.

## Results

**Membrane patterning of FtsA by treadmilling filaments of FtsZ.** FtsA localization to the division septum during the cell cycle is known to be FtsZ-dependent[9]. FtsZ also determines the circumferential dynamics of FtsA during treadmilling[10]. To study how FtsZ filaments direct positioning of FtsA on the membrane, we used a previously established in vitro reconstitution assay[4,21] based on dual-colour TIRF imaging of proteins binding to a glass supported lipid bilayer (Fig. S1a). Using this approach, we were able to simultaneously record the dynamics of fluorescently labelled FtsZ and FtsA on the membrane surface at high spatio-temporal resolution.

When we added fluorescently labelled FtsZ and FtsA to the supported membrane at a concentration ratio similar to the one found in vivo[22] (5:1) and lower (i.e. FtsZ = 1.25 μM and FtsA = 0.2 μM or 0.1 μM, with 75% Alexa488-FtsZ and 66% Cy5-FtsA) the proteins immediately formed a dynamic cytoskeleton pattern of treadmilling filaments where both proteins closely overlapped (Fig. 1c). We quantified the colocalization of the two fluorescent signals at steady state (after about 15 min incubation), where we obtained a high Pearson correlation coefficient (PCC) of 0.78 ± 0.04 (s.d. = standard deviation) and 0.69 ± 0.06 for 0.1 μM and 0.2 μM FtsA respectively (Fig. 1e). We were then wondering how this colocalization would be affected at higher FtsA concentrations. If its localization on the membrane was strictly FtsZ-dependent, we should observe strong colocalization of the two proteins, while excess protein would remain in solution. However, when we increased the bulk concentration of FtsA to 0.4 and 0.8 μM, we found that the FtsA pattern abruptly changed to show a homogeneous fluorescence on the membrane. As we still observed bundles of treadmilling FtsZ filaments, the corresponding colocalization coefficient dropped to PCC values of 0.32 ± 0.12 and 0.27 ± 0.09, respectively (Figs. 1c, e, and Supplementary Movie 1). This observation suggests that at high concentrations, FtsA binds to the membrane independently of FtsZ filaments. The appearance of the FtsZ pattern slightly changed with increasing FtsA concentrations, as well as the corresponding filament reorganization dynamics, which became slightly accelerated (Fig. 1i, S1b) as quantified by a faster decay of the temporal autocorrelation function[23]. However, we could not observe a significant effect on FtsZ treadmilling velocity or filament turnover (Fig. S1c, d).

Next, we wanted to quantify FtsA-FtsZ co-treadmilling dynamics, i.e. how efficiently FtsZ and FtsA recruit each other to the membrane during filament growth. For this aim, we prepared differential time lapse movies[24], where we subtract the

intensities of consecutive frames to selectively visualize the growing ends of filament bundles (Fig. 1d). We then calculated the Pearson correlation coefficient between the two channels of the differential movies ($PCC_{diff}$), which quantifies the covariation of the fluorescence signals for FtsA and FtsZ at the growing end of a filament bundle with a time resolution of the acquisition rate[4,23,24]. Like the colocalization coefficient (PCC), we found $PCC_{diff}$ to rapidly drop with increasing FtsA:FtsZ ratio, indicating that the ability of FtsZ to dynamically pattern FtsA assemblies on the membrane is severely compromised at high bulk concentrations of FtsA (Figs. 1d, f, and Supplementary Movie 2). In vivo, this property could contribute to the toxicity of FtsA observed at high expression levels as downstream cell division proteins would bind to FtsA independent of the Z-ring[20,25–27].

To understand how the activity state of FtsA affects colocalization with FtsZ filaments, we repeated these experiments with FtsA R286W, a well-known ZipA suppressor mutant with decreased self-interaction that is considered to represent an active form of the protein[19]. In contrast to the wild-type protein, we found that this mutant showed more robust colocalization with FtsZ, with high PCC values of around 0.8 at all concentration tested ($0.32 \pm 0.12$ vs $0.84 \pm 0.05$ for $0.4\,\mu M$ WT/R286W respectively, p-value: $1.28 \times 10^{-5}$; Fig. 1e, g, and Supplementary Movie 3). We also found FtsA R286W to co-migrate more efficiently with FtsZ filaments with constantly high $PCC_{diff}$ values ($0.18 \pm 0.04$ vs $0.76 \pm 0.04$ for $0.4\,\mu M$ WT/R286W respectively, p-value: $1.82 \times 10^{-4}$; Fig. 1f, h and Supplementary Movie 4). At the same time, we could not detect a difference in FtsZ treadmilling velocity, but slightly decreased turnover and increased fluorescence intensity of FtsZ compared to the wild-type protein (Fig. S1c–f). We also found the temporal autocorrelation function decayed more slowly for the filament pattern with FtsA R286W (Fig. 1i, Fig. S1b), suggesting that this mutant restricts filament reorganization. Together, these results corroborate earlier in vivo observations that FtsA R286W stabilizes the Z-ring[18] and demonstrate that this mutant outperforms wild-type FtsA in reproducing the spatiotemporal dynamics of treadmilling FtsZ filaments.

We found that wild-type FtsA and FtsA R286W strongly differ in their ability to localize to the FtsZ filament pattern (Fig. 1c, g). FtsZ interacts with FtsA via a highly conserved C-terminal peptide (CTP)[28], whose binding site on FtsA is located in its 2B subdomain, close to the Arginine residue mutated in FtsA R286W[11]. Concurrently, previous yeast two-hybrid experiments suggested that FtsA R286W has a higher affinity for FtsZ than the wild-type protein[13,19]. To test if an enhanced interaction with the CTP of FtsZ could explain the better colocalization of FtsA R286W with FtsZ filaments, we measured the binding time of a fluorescently labelled C-terminal FtsZ peptide (TAMRA-KEP-DYLDIPAFLRKQAD = TAMRA-CTP) with His-tagged versions of FtsA (FtsA-His6 and FtsA R286W-His6, also see Fig. 4) permanently attached to membranes containing dioctadecylamine (DODA)-tris-NTA, a $Ni^{2+}$-chelating lipid (Fig. S1g). For both versions of FtsA, we found only very transient recruitment of the membrane-bound peptide, with a mean life time of only $90 \pm 23\,ms$ for FtsA WT and $74 \pm 6\,ms$ for FtsA R286W (p-value 0.40; Fig. 1j, S1h) showing that the mutant protein tends to have weaker binding to the FtsZ CTP. A higher affinity of FtsA R286W towards FtsZ monomers seems, therefore, unlikely to be the reason for its increased colocalization with FtsZ filaments.

Another explanation for the loss of colocalization could be a higher membrane affinity of FtsA WT that results in indiscriminate membrane binding independent of FtsZ. In contrast, an active, monomeric FtsA with low membrane affinity would only be recruited to the membrane in a high avidity complex with FtsZ filaments and predominantly detach from the membrane if not

bound to FtsZ. To quantify the intrinsic membrane-affinity of the two versions of FtsA, we used Quartz Crystal Microbalance with Dissipation (QCM-D) and measured the hydrated mass of adsorbed protein on a membrane surface. We found that the membrane affinity of FtsA R286W was only slightly lower than that of FtsA WT ($K_d$ $0.32 \pm 0.05\,\mu M$ and of $0.21 \pm 0.01\,\mu M$, respectively) and that the amount of membrane-bound protein saturated at $0.8\,\mu M$ for both proteins (Fig. 1k and Fig. S1j). This result is consistent with the observation that FtsA WT and FtsA R286W behave identical in co-sedimentation experiments[12]. The small difference in membrane-binding affinities to the membrane cannot explain the observed contrast in colocalization of the two version of FtsA with FtsZ filaments in particular at high bulk concentrations of the proteins.

Electron microscopy experiments revealed that wild-type FtsA has the tendency to form membrane-bound arrays of minirings with a diameter of around 20 nm. In contrast, FtsA R286W was found to assemble into tightly packed short filaments and arcs[12]. While fluorescence microscopy cannot resolve molecular assemblies of these small dimensions, a higher packing density of FtsA R286W below FtsZ filaments could explain the enhanced colocalization with FtsZ. To test this hypothesis, we analysed the pixel-by-pixel relationship between the fluorescence intensities of FtsZ and FtsA in dual-colour fluorescence time-lapse movies. The linear slope of this relationship shows how the density of FtsA on the membrane changes when the FtsZ filament density increases and therefore is an indicator for the binding capacity of FtsZ filaments for FtsA (Fig. S1i). At low FtsA concentrations, we found the slopes for both proteins to be similar. Above $0.2\,\mu M$ these values dropped significantly for FtsA WT, but remained constant for FtsA R286W at ~0.7 ($0.23 \pm 0.18$ vs $0.72 \pm 0.16$ for $0.4\,\mu M$ WT/R286W respectively, p-value: $1.22 \times 10^{-2}$) (Fig. 1l, S1i). These results indicate that at high concentrations the amount of FtsA WT that can be recruited to FtsZ filaments is strictly limited, likely due to the formation of minirings, while FtsA R286W continues to accumulate on the FtsZ filament.

**FtsA R286W allows for enhanced recruitment of $FtsN_{cyto}$ to FtsZ filaments.** FtsA provides a physical link between treadmilling FtsZ filaments in the cytoplasm and cell division proteins located in the membrane. However, polymerization of FtsA and recruitment of downstream proteins are thought to be mutually exclusive as they both involve interactions via FtsA's 1C domain[29]. Accordingly, in an active, depolymerized FtsA this domain would be readily available to interact with downstream proteins like FtsN. Vice versa, binding of these proteins should facilitate the transition of FtsA from the inactive, oligomeric to the active, more monomeric form.

To test these predictions, we mimicked the presence of transmembrane FtsN in the bilayer by attaching its His-tagged, cytoplasmic peptide ($FtsN^{1-32}$-His6x = $FtsN_{cyto}$) to the surface of a supported membrane containing 0.25% Tris-NTA lipids[4]. We then used these modified membranes to compare how the two versions of FtsA differ in their ability to recruit $FtsN_{cyto}$ to treadmilling filaments of FtsZ (Fig. 2a). Closely mirroring the behaviour observed for the localizations of FtsA (Fig. S2a, b), we found strong overlap (PCC) and co-treadmilling ($PCC_{diff}$) of $FtsN_{cyto}$ with FtsZ filaments at low concentrations of FtsA WT ([FtsA WT] < $0.4\,\mu M$) and a sudden drop of these values at higher concentrations (Fig. 2b–d, and Supplementary Movies 5, 7). In contrast, in the case of FtsA R286W, both values remained > 0.6, even at concentrations above $0.4\,\mu M$ (PCC: $0.30 \pm 0.05$ vs $0.70 \pm 0.02$, p-value: $4.92 \times 10^{-4}$ and $PCC_{diff}$: $0.17 \pm 0.04$ vs $0.47 \pm 0.05$ for $0.4\,\mu M$ WT/R286W respectively, p-value: $3.79 \times 10^{-3}$; Fig. 2c–e, and Supplementary Movie 6, 8).

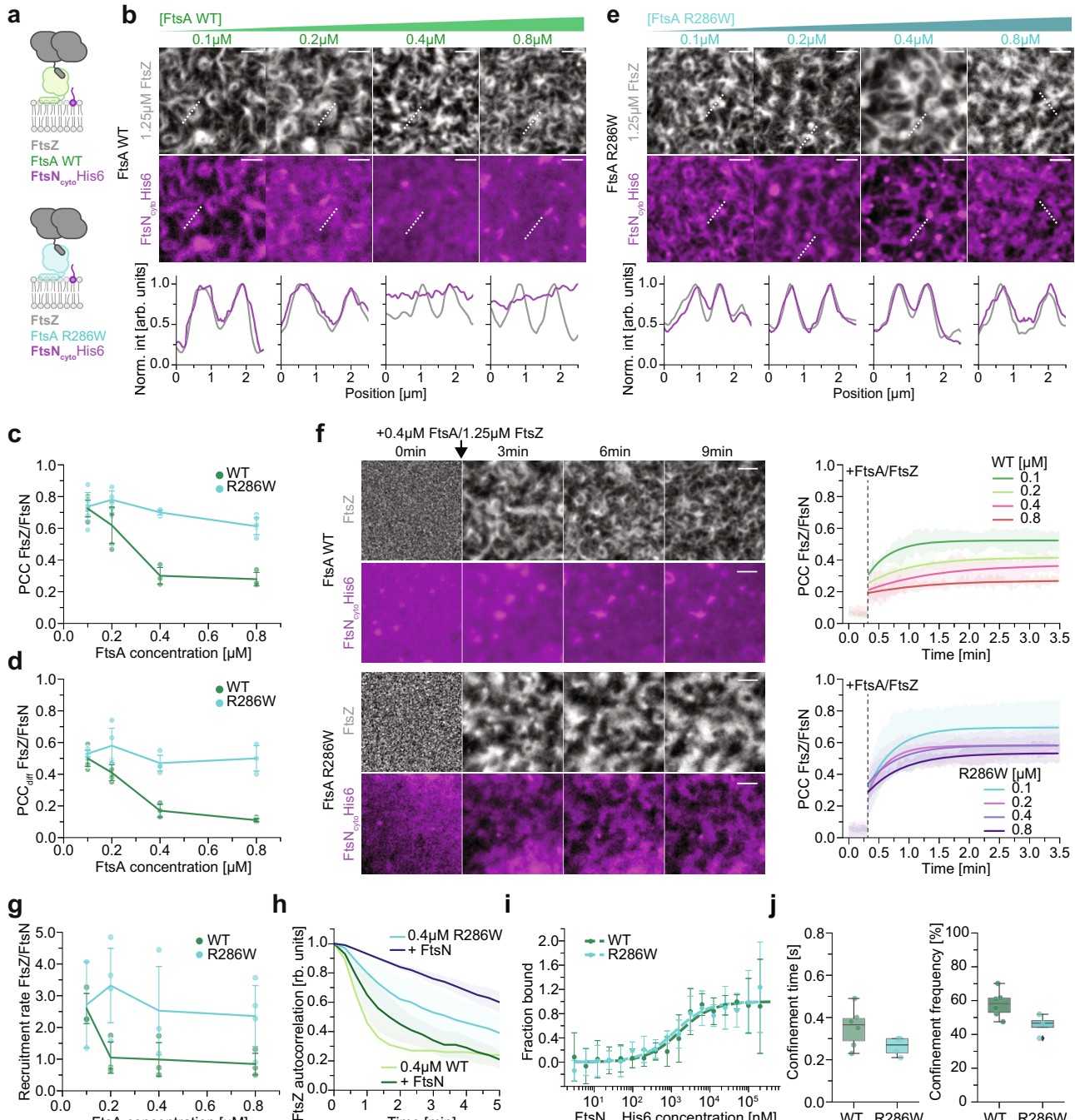

**Fig. 2 FtsA R286W shows enhanced recruitment of FtsN$_{cyto}$ to FtsZ filaments. a** Schematic illustrating added components and fluorescently labelled proteins (bold) in experiments shown in **b** and **e**. **b** Representative micrographs of Alexa488-FtsZ (grey) and Cy5-FtsN$_{cyto}$ (magenta) at increasing FtsA WT and constant FtsZ concentrations. Intensity profiles correspond to the dashed white lines. **c** Colocalization of FtsZ and FtsN$_{cyto}$ in the presence of FtsA WT (green) or FtsA R286W (cyan) quantified by PCC. $n$(WT) = 5/5/3/3 and $n$(R286W) = 6/4/4/5. **d** Dynamic colocalization of FtsZ and FtsN$_{cyto}$ in the presence of FtsA WT (green) or FtsA R286W (cyan) quantified by PCC$_{diff}$. $n$(WT) = 5/5/3/3 and $n$(R286W);= 4/4/4/3. **e** Representative micrographs of Alexa488-FtsZ (grey) and Cy5-FtsN$_{cyto}$ (magenta) at increasing FtsA R286W and constant FtsZ concentration. Line profiles were taken along dashed white lines. **f** Top: Cy5-FtsN$_{cyto}$ (magenta) is not enriched on FtsZ filaments after adding Alexa488-FtsZ (grey) and 0.4 μM FtsA WT at 0 min. Bottom: In experiments with 0.4 μM FtsA R286W, Cy5-FtsN$_{cyto}$ colocalizes well with FtsZ filaments. Right: Increase of PCC after addition of FtsZ and different FtsA WT (top) or R286W (bottom) concentrations. Thick lines depict power law exponential fits towards the mean PCC, shaded area is the standard deviation. **g** Quantification of the recruitment rate of FtsN$_{cyto}$ towards FtsZ filaments, extracted from experiments in **f** by fitting a power law exponential to individual experiments. $n$(WT) = 3/3/3/3 and $n$(R286W) = 2/4/3/4. **h** FtsN$_{cyto}$ increases the persistency of the FtsZ network with both, FtsA WT and R286W. $n$(WT/R286W) = 3. **i** Quantification of the binding affinity of FtsN$_{cyto}$ towards FtsA WT or R286W by MST. $n$(WT, R286W) = 3. **j** Quantification of the duration and frequency of FtsN$_{cyto}$ single molecule confinement events. $n$(WT/R286W) = 6/4. Scale bars are 2 μm. Dots in all plots represent independent experiments, thick lines indicate the mean and error bars or shaded areas depict the standard deviation. The boxes indicate the 25–75th percentiles, whiskers show the maximum/minimum values within the standard deviation, the midline indicates the median and diamonds indicate outliers. Source data are provided as a Source Data file.

Next, we were wondering about the rate of FtsN$_{cyto}$ accumulation on FtsZ-FtsA co-filaments. Starting from a homogeneous distribution of the membrane-bound peptide, we measured how quickly the overlap of the FtsN$_{cyto}$ and FtsZ signals increased after adding FtsA and FtsZ (Fig. 2f, Fig. S2c, d). By fitting an exponential function to the increase of the PCC with time, we were able to extract the corresponding recruitment rate (Fig. 2f, g). While FtsN$_{cyto}$ enrichment saturated within 1 min for all concentrations of FtsA R286W, the recruitment rate for FtsA WT dropped significantly already at 0.2 µM FtsA, such that it required more than twice as long for FtsN$_{cyto}$ to colocalize with FtsZ. Together, these data demonstrate FtsA R286W recruits downstream proteins to FtsZ filaments more efficiently than wild-type FtsA. This property could also explain why cell division is faster in cells with FtsA R286W[18].

Previous literature suggested that arrival of FtsN at the Z-ring triggers disassembly of FtsA WT oligomers into a more FtsA R286W-like, monomeric state[7]. Thus, we were wondering if the presence of FtsN$_{cyto}$ could change the colocalization of FtsA WT with FtsZ to resemble the behaviour of FtsA R286W. While addition of FtsN$_{cyto}$ slightly increases the total amount of both proteins on the membrane, their overlap and the density of FtsA on FtsZ filaments, it could not prevent the loss of colocalization at higher concentrations of FtsA (Fig. S2e–h). This shows that the presence of FtsN$_{cyto}$ has no strong effect on the recruitment of FtsA towards FtsZ filaments and that even with FtsN$_{cyto}$, FtsA WT does not resemble FtsA R286W. However, we found that adding FtsN$_{cyto}$ to either version of FtsA significantly slows down the reorganization dynamics of the FtsZ pattern, suggesting that binding of FtsN$_{cyto}$ leads to a transition in both FtsAs that prevents FtsZ filament realignment (Fig. 2h, Fig. S2i).

Next, we wanted to know if the increased overlap between FtsN$_{cyto}$ and FtsZ with FtsA R286W can be explained by an increased affinity of FtsN$_{cyto}$ towards the hypermorphic mutant as suggested previously[5,19]. To test this idea, we performed microscale thermophoresis (MST) experiments with fluorescently labelled FtsA WT and FtsA R286W and increasing concentrations of FtsN$_{cyto}$ (Fig. 2i and Fig. S2j). For both proteins, we measured similar dissociation constants in these experiments of $K_D$(Cy5-FtsA WT/FtsN$_{cyto}$) = 1.58 ± 0.43 µM and $K_D$(Cy5-FtsA R286W/FtsN$_{cyto}$) = 1.17 ± 0.37 µM. Since FtsN interacts with FtsA on the membrane surface, we were wondering if the two-dimensional confinement could enhance the difference. We, therefore, imaged the trajectories of individual membrane-bound FtsN$_{cyto}$ peptides in the presence of treadmilling FtsZ-FtsA filaments and then quantified the duration and frequency of their transient confinement[4] (Fig. 2j and Fig. S2k, l and Supplementary Movie 9). We found that both values were in fact slightly lower for FtsA R286W, confirming that it does not have increased affinity towards FtsN$_{cyto}$ (0.35 ± 0.09 s vs 0.26 ± 0.04 s, $p$-value: 0.11 and 57.99 ± 7.29% vs 45.70 ± 5.10%, $p$-value: 0.03).

We conclude that our in vitro experiments recapitulate several observations made in the living cell, where FtsA R286W recruit FtsN to FtsZ filaments more efficiently and its arrival at midcell stabilizes the Z-ring. However, we found this difference is not due to an enhanced affinity of this peptide to FtsA R286W, but likely the result of the higher packing density of FtsA R286W below FtsZ filaments (Fig. 1l).

**FtsN$_{cyto}$ does not depolymerize FtsA WT oligomers and enhances interaction of FtsA R286W.** So far, we have seen that FtsA R286W is more strongly recruited to FtsZ filaments (Fig. 1), which allows for an improved recruitment of FtsN$_{cyto}$ towards FtsA-FtsZ co-filaments (Fig. 2). Additionally, we also found that FtsA R286W co-migrates with treadmilling FtsZ filaments more

efficiently (Fig. 1d, f, h). As FtsA WT and FtsA R286W do not significantly differ in their affinities towards FtsZ (Fig. 1j), the membrane (Fig. 1k) or FtsN$_{cyto}$ (Fig. 2i, j), we decided to investigate a possible mechanism that could explain our observations.

First, we studied the behaviour of single FtsA proteins on the membrane surface. In a background of unlabelled proteins, we followed individual FtsA WT and FtsA R286W proteins and analysed their trajectories by single-particle tracking (Fig. 3a). At 0.1 µM, we found that FtsA WT showed a low mobility with a diffusion coefficient of 0.14 ± 0.04 µm²/s and a mean residence time on the membrane of 19.4 ± 3.9 s. With increasing bulk concentrations and protein densities, the membrane residence time increased (35.1 ± 2.0 s at 0.8 µM FtsA), while the diffusion coefficient dropped to a value of 0.004 ± 0.001 µm²/s indicating almost immobile proteins on the membrane. Similar to previous FRAP experiments in vivo[18], we found a faster exchange of FtsA R286W with a single molecule residence time 2–7 fold shorter than that of wild-type FtsA (from 2.6 ± 0.5 s to 14.1 ± 2.3 s for 0.1 and 0.8 µM respectively). The diffusivity of FtsA R286W also decreased at higher protein concentrations (from 0.42 ± 0.06 µm²/s to 0.05 ± 0.01 µm²/s for 0.1 and 0.8 µM respectively), but remained mobile (Fig. 3b and Supplementary Movie 10). These differences in residence time and diffusion likely correlate with different modes of self-interaction of the two proteins[12,19].

To directly measure FtsA self-interaction in our fluorescence microscopy experiments, we established a FRET (Förster resonance energy transfer)-based assay using FtsA labelled with either Cy5 (acceptor) or Cy3 (donor). First, we tried to find evidence for self-interaction in solution, but could not detect any significant FRET signal under these conditions or oligomerization in SEC-MALS experiments (Fig. S3a, S3b) demonstrating that full-length FtsA is monomeric in solution. However, we found that FRET increased significantly in the presence of lipid vesicles, indicating that FtsA can oligomerize on the lipid membrane (Fig. S3b, S3c). We then decided to use this approach to measure the degree of FtsA self-interaction on supported lipid membranes by quantifying the increase in donor fluorescence after photobleaching the acceptor fluorophore[30–32] (Fig. 3c–e and Fig. S3d). Without acceptor fluorophore, we saw no change in donor intensity after photobleaching (Fig. S3e). As a negative control, we also attached the Hexahistidine-SUMO-tag (His-SUMO) labelled with Cy5 and Cy3 to membranes with an increasing density of Tris-NTA lipids. For this negative control, we did not find any significant FRET, even at maximal coverage of the membrane surface (Fig. 4g).

We found significant FRET for both versions of FtsA, while the efficiency for FtsA WT was generally higher than for R286W (22.1 ± 5.7% vs 8.1 ± 2.8% for 0.4 µM WT/R286W respectively, $p$-value: 6.12 × 10⁻⁹; Figs. 3f–h and Supplementary Movies 11, 12), and that it increased with its bulk concentration indicative for stronger FtsA WT self-interaction. We could also use the data from these bleaching experiments to quantify the membrane-binding kinetics and diffusion of membrane-bound proteins by analysing the change of the fluorescence profile during recovery[33]. In agreement with our single-molecule experiments, we found a much faster exchange (0.04 ± 0.01 s⁻¹ vs 0.37 ± 0.24 s⁻¹ for 0.4 µM WT/R286W respectively, $p$-value: 6.73 × 10⁻⁶, Fig. 3i) and diffusion for FtsA R286W compared to the wild-type protein (0.01 ± 0.01 µm²/s vs 0.14 ± 0.08 µm²/s for 0.4 µM WT/R286W respectively, $p$-value: 1.05 × 10⁻⁶, Fig. 3j). This fast turnover of FtsA R286W on the membrane can also explain the shorter confinement time we found for FtsN$_{cyto}$ (Fig. 2j). Consistent with our QCM-D experiments (Fig. 1k), we found that FRET and membrane exchange plateaued at around 50% at FtsA concentrations above 0.8 µM (Fig. S3i–S3k). Furthermore, we found that

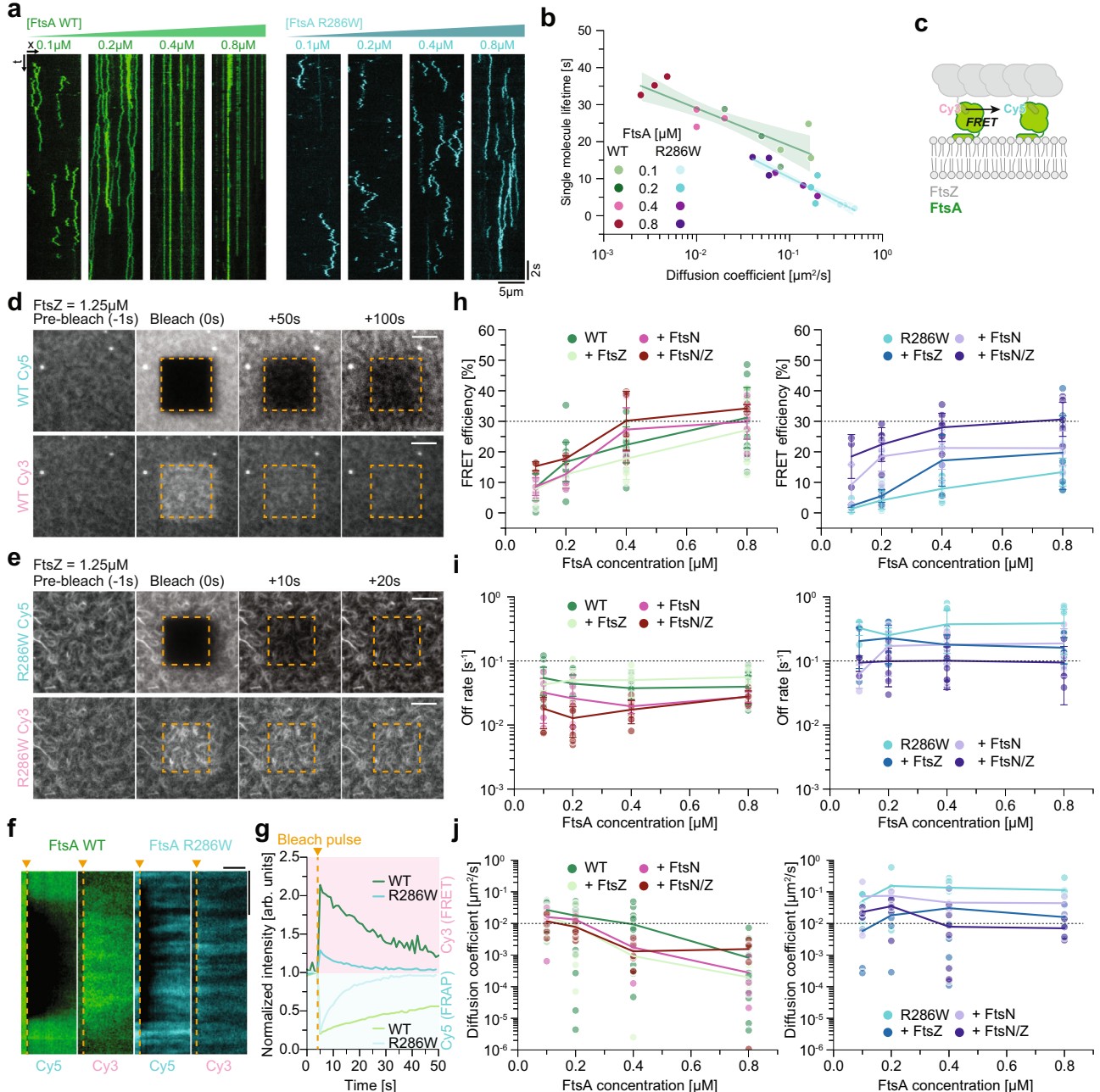

**Fig. 3 FtsA R286W shows faster membrane exchange than FtsA WT, while the self-interaction of both proteins is enhanced in the presence of FtsN_cyto. a** Representative single-molecule kymographs of Cy5-FtsA WT (green) and Cy5-FtsA R286W (cyan) at increasing concentrations. **b** With higher concentrations the mobility of both proteins decreases and the residence time increases, but R286W remains 10x more mobile and has a 2-7x lower lifetime. Thick line indicates the mean value, shaded area represents 95% confidence interval; $n = 3$. **c** Schematic of the experiment to measure FRET between Cy3-FtsA (donor) and Cy5-FtsA (acceptor). **d**, **e** Representative montage of acceptor photobleaching experiment with FtsA WT (**d**) or R286W (**e**). Scale bars are 5 μm. **f** Kymographs from acceptor photobleaching experiments, and corresponding donor signal for FtsA WT (green) and R286W (cyan). Scale bars are 4 μm and 20 s, respectively. **g** Representative curves showing acceptor (top) and donor fluorescence intensities (bottom) for FtsA WT (green) and R286W (cyan). **h** Left: FtsA WT FRET increases at higher concentrations, FtsZ slightly decreases, while FtsN_cyto and FtsZ/FtsN_cyto increase FtsA self-interaction. Right: FtsA R286W FRET increases with concentration, but is significantly lower than for WT. FtsZ and FtsN_cyto increase FRET, and with both the FRET signal of WT and R286W are indistinguishable. **i** Left: Off-rates remain slow with increasing concentrations of WT. FtsN_cyto and FtsN_cyto/FtsZ further decrease the exchange. Right: Off-binding rates remain constantly high with increasing concentrations of FtsA R286W. FtsN_cyto and FtsN_cyto/FtsZ decrease the rate, but it remains higher than for WT. **j** Left: $D_{coeff}$ drops with increasing FtsA WT concentrations and decreases further with FtsZ and FtsN_cyto. Right: $D_{coeff}$ remains unchanged at increased concentrations of FtsA R286W. FtsN_cyto and FtsZ slow down R286W, which remains more mobile than WT. Dots in all plots represent independent experiments, thick lines indicate the mean and error bars depict the standard deviation. Source data are provided as a Source Data file.

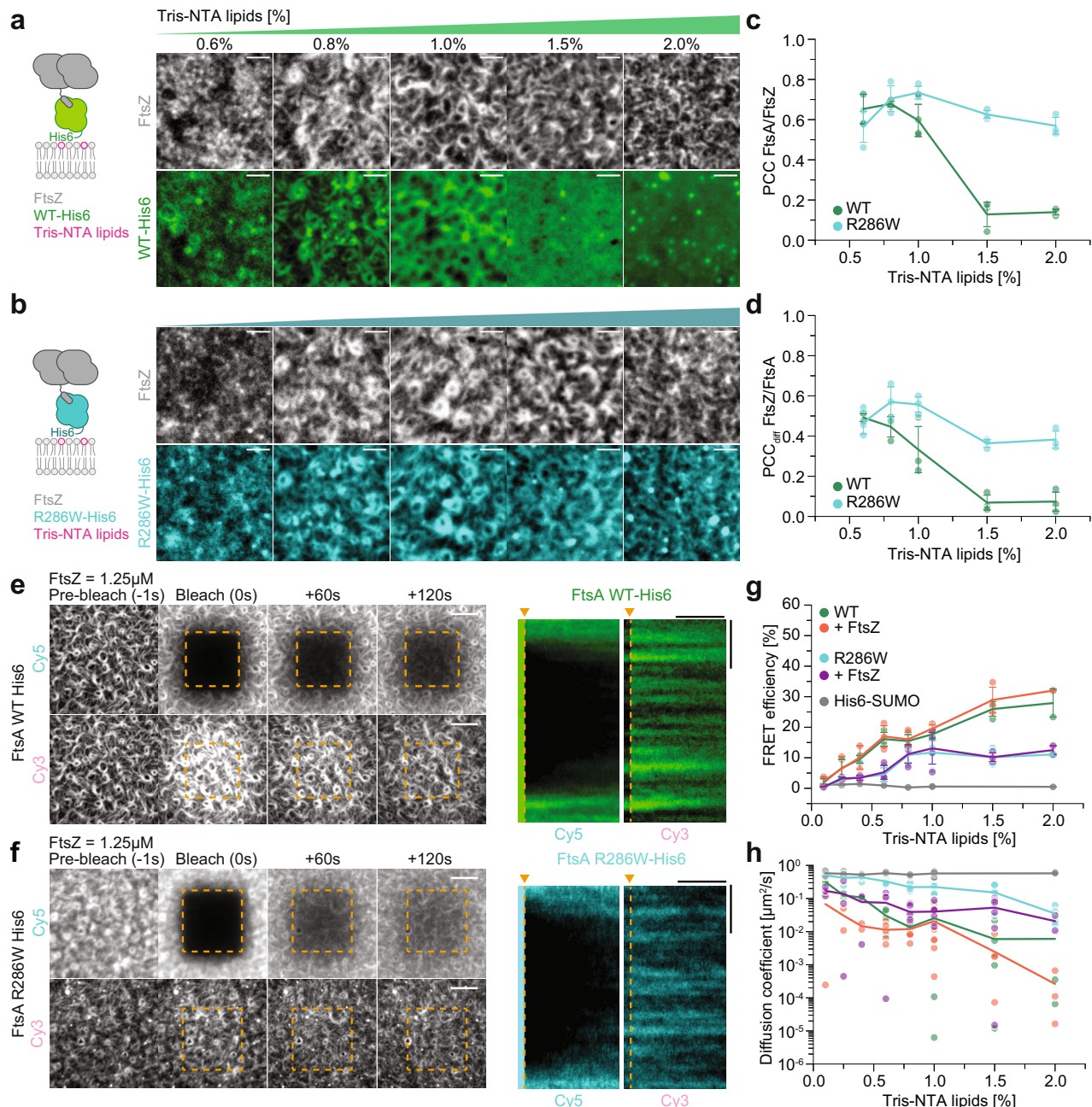

**Fig. 4 Membrane diffusion and protein binding dynamics contribute to co-treadmilling of FtsA and FtsZ. a, b** Representative micrographs of Alexa488-FtsZ (grey) and Cy5-FtsA WT-His6 (green) (**a**) or Alexa488-FtsZ (grey) and Cy5-FtsA R286W-His6 (cyan) (**b**) at increasing Tris-NTA lipid densities. Scale bars are 2 μm. **c** Colocalization of FtsZ with WT-His6 (green) or R286W-His6 (cyan) quantified by PCC. **d** Dynamic colocalization of FtsZ with WT-His6 (green) or R286W-His6 (cyan) quantified by PCC$_{diff}$. The experiments to measure PCC and PCC$_{diff}$ were repeated three times for each Tris-NTA lipid density. **e, f** Representative micrographs of acceptor bleaching recovery and donor intensity increase of FtsA WT-His6 (**e**) and FtsA R286W-His6 (**f**). Scale bars are 5 μm. Right: Kymographs depicting the bleaching recovery, as well as the FRET signal for FtsA WT-His6 (green) or R286W-His6 (cyan). Scale bars are 4 μm and 40 s, respectively. **g** While self-interaction of His-tagged FtsAs increases with increasing protein density, there was no FRET for His6-SUMO detected. FRET signal for FtsA WT-His6 is higher compared to FtsA R286W-His6 and FtsZ has no effect on the self-interaction. **h** While the mobility of His6-SUMO remains unchanged, the diffusion coefficient of His-tagged FtsAs decreases with increasing protein density. The experiments in **g** and **h** were replicated twice for Tris-NTA densities < 0.6% and three times for Tris-NTA densities > 0.6%. Dots in all plots represent independent experiments, thick lines indicate the mean and error bars depict the standard deviation. Source data are provided as a Source Data file.

replacing ATP with a non-hydrolysable analogue ATPγS had no significant effect on membrane-binding, self-interaction or membrane-binding dynamics (Fig. S3l–S3n). Together, these data confirm that self-interaction of membrane-bound FtsA WT is enhanced, which results in slower exchange dynamics compared to the mutant protein.

Next, we wanted to find out how binding of FtsZ and FtsN$_{cyto}$ affect FtsA protein exchange, diffusion and self-interaction on the membrane. In the presence of FtsZ, FtsA R286W still exchanged one order of magnitude faster than the wild-type protein, with membrane off-rates of around 0.20 s$^{-1}$ and 0.05 s$^{-1}$, respectively (Fig. 3i, S3g). As the off-rates for FtsZ are between 0.10 and 0.20 s$^{-1}$

(Fig. S1d), this means that FtsA R286W turns over 1–2 times faster than FtsZ monomers in the treadmilling filament, while FtsA WT remains bound about 2–4 times longer. For both versions of FtsA, the diffusion coefficient was decreased in the presence of FtsZ (Fig. 3j, S3h).

Next, we were interested in testing how FtsN$_{cyto}$ promotes deoligomerization of FtsA WT and reduces the corresponding FRET efficiency[7]. As suggested by the current model of divisome activation, we expected a strong decrease in FRET for FtsA WT, while it should stay unchanged or be reduced only slightly for FtsA R286W. Surprisingly, we found that for FtsA WT the presence of FtsN$_{cyto}$ resulted in a small FRET increase. In the case of FtsA R286W, the FRET efficiency was even increased three-fold when both binding partners, FtsN$_{cyto}$ and FtsZ were present, up to the same level as found for FtsA WT (30.2 ± 9.6% vs 28.1 ± 4.6% for 0.4 μM WT/R286W respectively, p-value: 0.64) (Fig. 3h, Fig. S3f and Supplementary Movie 13). In addition, the presence of FtsN$_{cyto}$ slightly decreased the detachment rate and membrane mobility of both FtsA WT and FtsA R286W (Fig. 3i-j, Fig. S3g-h and Supplementary Table 1). Together, these observations show that in contrast to previous reports, FtsN$_{cyto}$ does not disassemble FtsA oligomers[3,16,19]. Instead, our data suggests that FtsN$_{cyto}$ supports the formation of FtsA polymers, possibly to due enhanced lateral interactions[12], resulting in a high FRET efficiency for both versions of FtsA.

**Lateral diffusion and membrane-binding dynamics contribute to co-treadmilling of FtsA with treadmilling FtsZ filaments.** Co-treadmilling of FtsA with FtsZ relies on the dynamic exchange of FtsA on the membrane. We have found that FtsA R286W shows a dramatically shorter residence time and decreased self-interaction compared to the wild-type protein (Fig. 3). These two properties strongly correlate with a much stronger colocalization with treadmilling FtsZ filaments (Fig. 1). To investigate the respective contributions of membrane binding kinetics and FtsA self-interaction on the colocalization dynamics with FtsZ, we wanted to create variants of the two FtsA proteins that are permanently attached to the membrane. We, therefore, replaced their amphipathic helices by C-terminal His-tags to obtain FtsA WT (1-405)−6xHis (=FtsA-His6) and FtsA R286W (1-405)−6xHis (=FtsA R286W-His6), which we attached to membranes containing different amounts of Tris-NTA lipids (Fig. 4a, b). Accordingly, in these experiments the proteins only differ in their tendency to form oligomers.

When we measured the colocalization of FtsZ with the His-tagged versions of FtsA, we found that just like for the native proteins, colocalization and co-treadmilling decreased with increasing FtsA-His6 densities on the membrane, while they stayed almost constant for FtsA R286W-His6 (PCC: 0.13 ± 0.06 vs 0.63 ± 0.02 and PCC$_{diff}$: 0.07 ± 0.04 vs 0.36 ± 0.02 at 1.5% Tris-NTA for WT/R286W-His6 respectively, p-values: 3.86 × 10$^{-4}$ and 5.70 × 10$^{-4}$; Fig. 4a–d and Supplementary Movie 14, 15). Furthermore, despite identical densities on the membrane at a given amount of Tris-NTA lipids, colocalization of FtsA R286W-His6 with FtsZ was always higher than for FtsA-His6, confirming that FtsZ filaments have a higher capacity for FtsA R286W than for the wildtype protein[12] (Fig. 1l). Interestingly, co-treadmilling with FtsZ was about 50% reduced for FtsA R286W-His6 compared to the reversibly membrane-binding protein (0.76 ± 0.05 vs 0.36 ± 0.02 for 0.8 μM R286W and R286W-His6 on 1.5% Tris-NTA respectively, p-value: 7.78 × 10$^{-5}$; Fig. 1f, Fig. 4d), emphasizing that although the fast diffusion of FtsA R286W-His6 allows for some degree of co-treadmilling with FtsZ filaments, recruitment of the protein from solution significantly contributes to its efficiency.

To evaluate the degree of FtsA self-interaction in the absence of protein exchange, we measured the FRET efficiency and diffusion coefficient of FtsA-His6 and FtsA R286W-His6 at different Tris-NTA densities. Similar to the proteins with native membrane binding, permanently attached FtsA R286W showed lower FRET efficiency and faster diffusion at all densities tested, showing that faster membrane-binding dynamics are the consequence of reduced self-interaction of FtsA R286W-His6 and not vice versa. (Fig. 4e–h, Fig. S4j-S4l and Supplementary Movie 16).

While the FRET efficiency for FtsA R286W-His6 was not increased in the presence of FtsZ filaments, we found two-times higher FRET when both FtsN$_{cyto}$ and FtsA R286W-His6 were attached to the membrane (10.19 ± 1.44% vs 17.73 ± 3.06%, p-value: 0.03; Fig. S5e, f). At the same time, we found that under these conditions the proteins were practically immobile on the membrane and the corresponding pattern dramatically perturbed (0.05 ± 0.04 μm²/s vs 8.43 × 10$^{-4}$ ± 1.17 × 10$^{-3}$ μm²/s, p-value: 0.01; Fig. S5a–d, and Fig. S5g, h and Supplementary Movies 17, 18). Together, we can conclude that the different FRET efficiencies and membrane mobilities of FtsA-His6 and FtsA R286W-His6 are solely due to their different tendencies to oligomerize and that FtsN$_{cyto}$ strongly enhances polymerization.

**A model for the behaviour of FtsA during divisome maturation.** Using a minimal set of purified components in combination with quantitative fluorescence microscopy, we confirmed previous conclusions on the properties of FtsA and its hyperactive mutant FtsA R286W made from in vivo observations. In in vitro experiments, we were also able to provide new insights into the role of FtsA during the assembly and activation of the bacterial cell division machinery. First, we demonstrate that FtsZ filaments define the spatiotemporal distribution of FtsA WT assemblies on the membrane[9] and that FtsA R286W stabilizes FtsZ filament reorganization compared to the wildtype protein[8] (Fig. 1). Second, we confirm that FtsA R286W outperforms FtsA WT in the recruitment of downstream proteins[19] (Fig. 2). Furthermore, our results from FRET and single-molecule experiments support previous findings on the oligomeric nature of FtsA as well as a decreased self-interaction in case of FtsA R286W[19] (Fig. 3). However, while we expected that binding of FtsN$_{cyto}$ disassembles FtsA WT oligomers, we found that FtsN$_{cyto}$ increases FtsA self-interaction, in particular for the intrinsically less-oligomeric mutant FtsA R286W.

Previous electron microscopy studies found that FtsA is able to form oligomers of different conformations, such as straight filaments and minirings, but also filament doublets, which form predominantly in the case of several ZipA suppressor mutants[11,12,34]. It therefore seems likely that FtsA can oligomerize via different interfaces allowing for lateral and longitudinal interactions[12]. As longitudinal interactions are compromised in the case of FtsA R286W[12,34], binding of FtsN$_{cyto}$ likely induces a conformational change that promotes lateral interactions, enhancing the formation of filament doublets and increasing the FRET efficiency (Fig. 3h, Fig. S3f, 5a). This FtsN-dependent structural transition of FtsA goes along with an enhanced recruitment of FtsZ filaments and a decrease of filament reorientation (Fig. 2h, S2f). Importantly, these interpretations are supported by a concurrent study that finds that FtsA switches from minirings to double filaments upon binding of FtsN[35]. Together, these data suggest that the active version of FtsA is not a monomer, but an FtsA double-filament that forms upon binding the cytoplasmic peptide of FtsN.

The concentration dependent tendency of FtsA WT to organize into ring arrays[12] also offers an explanation for the transition from co-migrating dynamic assemblies towards a

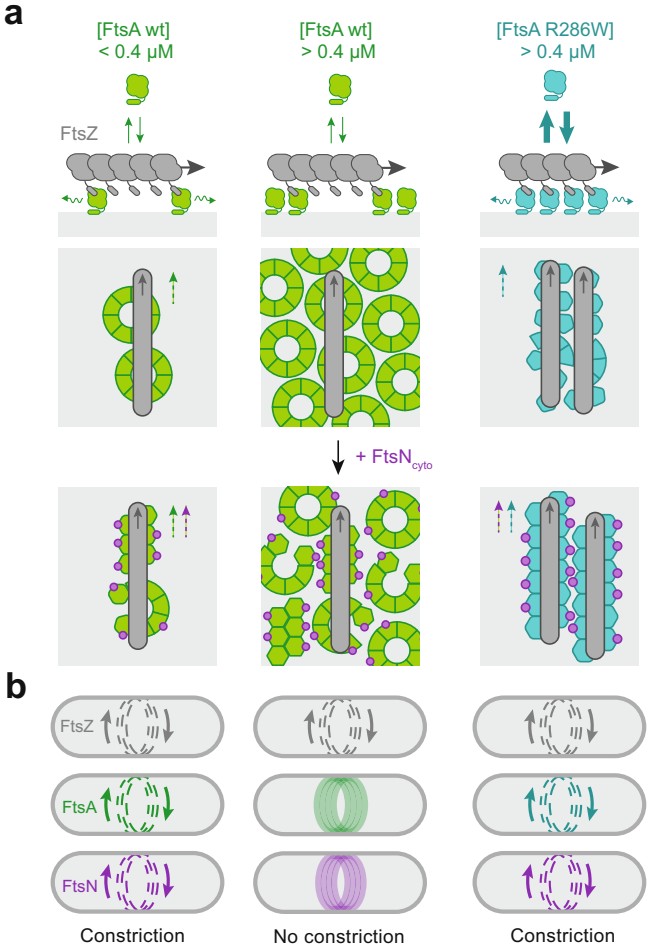

**Fig. 5 A model for the behaviour of FtsA during divisome activation.** The spatiotemporal distribution of FtsA is the result of its dynamic interaction with treadmilling FtsZ filaments and the lipid membrane. Grey arrows indicate FtsZ treadmilling direction, green and cyan arrows represent cycling exchange of FtsA by reversible membrane binding, wavy arrows represent diffusion on the membrane surface, dashed arrows indicate co-treadmilling of FtsA and FtsN with FtsZ. **a** Left: At low FtsA WT concentrations, i.e. at a protein ratio found in vivo ([FtsA WT] = 0.2 μM, [FtsZ] = 1.25 μM), diffusion of membrane-bound FtsA is fast enough to allow for co-treadmilling despite slow-cycling rates. The presence of FtsA WT minirings limits the packing density and therefore its amount recruited to FtsZ filaments. Furthermore, FtsA minirings allow for a continuous realignment of treadmilling filaments. Binding of FtsN$_{cyto}$ leads to the formation of FtsA-FtsN copolymers that follow treadmilling FtsZ filaments and limits their reorganization. Middle: At higher concentrations, FtsA WT forms a continuous array of minirings, stabilized by lateral interactions. Due to the absence of diffusion, FtsA cannot follow FtsZ treadmilling dynamics. Right: Even at high FtsA R286W concentrations, its fast exchange allows co-treadmilling with FtsZ filament. Loss of longitudinal interactions in FtsA R286W disrupts minirings and allows for a higher packing density, which limits the realignment of FtsZ filaments. Binding of FtsN further enhances lateral interactions to promote the formation of FtsA double filaments that further limits realignment of FtsZ filaments. **b** In vivo, low concentrations of FtsA WT allow for continuous co-treadmilling of FtsZ, FtsA and downstream proteins around the cell diameter, where they distribute cell wall synthesis allowing for cell constriction. When overexpressed, FtsA WT as well as its binding partner FtsN stop following FtsZ treadmilling, cell wall synthesis is not distributed and the cell fails to divide. In the case of FtsA R286W, its fast membrane cycling rate and faster diffusion allows for a dynamic distribution of proteins, constriction and cell division, even when overexpressed.

stable, homogeneous protein layer on the membrane (Figs. 1c, 3a, and 5a, b). Conversely, the absence of higher-order structures in case of FtsA R286W can account for its faster membrane exchange as well as its tighter packing below FtsZ filaments (Figs. 1g, 3a, and 5a). The increased density of FtsA R286W leads to an enhanced recruitment of FtsN$_{cyto}$ (Fig. 2) and possibly of other weakly binding proteins such as FtsQ[4,6,36] and FtsW[14]. We think this property alone could explain the ability of FtsA R286W to bypass otherwise essential cell division proteins in vivo[19].

Finally, our experiments also shed light on the general requirements for the signalling function of FtsA during cytokinesis, i.e. its ability to transmit the spatiotemporal information originating from treadmilling FtsZ filaments in the cytoplasm towards the periplasmic space. This function relies on a close replication of FtsZ polymerization dynamics, which is the result of the dynamic exchange of FtsA by lateral diffusion and via membrane binding and detachment. As both of these two processes are much faster for FtsA R286W than for FtsA WT (Fig. 3i, j), FtsA R286W can sample the dynamic filament with a minimal loss of spatiotemporal information[37]. We, therefore, expect FtsA R286W to not only be better at directing cell division to midcell, but also at homogenously distributing cell wall synthesis around the division site. It will be interesting to study the behaviour of single FtsA molecules in vivo and how different exchange dynamics of the membrane anchor correlate with the ability of treadmilling FtsZ filaments to drive the directional motion of cell division proteins[38].

With the described benefits of the R286W mutation for the roles of FtsA, the question arises why it did not persist during evolution. In vivo, FtsA R286W produces misaligned and twisted division septa and minicells. Accordingly, it is possible that the longer residence times of FtsA WT oligomers, a tighter control of their hypothesized structural reorganization, as well as the dependency on ZipA as a second membrane anchor provide additional control mechanisms that increase the precision and robustness of E. coli division. Since ZipA, as well as FtsN, are not as widely conserved as FtsA, it will also be interesting to study the mechanisms of divisome activation in other bacterial, i.e. Gram-positive species that lack FtsN and where FtsA is not essential for division.

## Methods

**Reagents.** All of the reagents, chemicals, peptides and software used are listed in Supplementary Table 2. The peptide sequence as well as protein parameters are listed in Supplementary Table 3.

**Purification and fluorescence labelling of FtsZ.** FtsZ was purified as previously described[4,21]. In short, FtsZ with N-terminal His$_6$-SUMO fusion protein and seven residues (AEGCGEL) for maleimide coupling of thiol-reactive dyes was cloned into a pTB146-derived vector[21]. FtsZ was expressed in E. coli BL21 cells, at 37 °C in Terrific Broth supplemented with 100 μg ml$^{-1}$ ampicillin and expression was induced at an OD600 of 0.6–0.8 with 1 mM isopropyl-β-thiogalactopyranoside (IPTG) and incubated for 5 h at 37 °C. Cells were harvested by centrifugation (5000 g for 30 min at 4 °C). The pellet was resuspended in buffer A (50 mM Tris-HCl [pH 8.0], 500 mM KCl, 2 mM β-mercaptoethanol and 10% glycerol) plus 20 mM imidazole and supplemented with ethylenediaminetetraacetic acid (EDTA)-free protease inhibitor cocktail tablets (Roche Diagnostics). Cells were lysed using a cell disrupter (Constant Systems; Cell TS 1.1) at a pressure of 1.36 kbar and subsequently incubated with 2.5 mM MgCl$_2$ and 1 mg ml$^{-1}$ DNase for 15 min. Cell debris was removed by centrifugation at 60,000 g for 30 min at 4 °C and the supernatant was incubated with nickel-nitrilotriacetic acid (Ni-NTA) resin (HisPur Ni-NTA Resin; Thermo Fisher Scientific) for 1 h at 4 °C. The resin was washed with buffer A containing 10 mM imidazole, followed by buffer A with 20 mM imidazole and the protein was subsequently eluted with buffer A supplemented with 250 mM imidazole. To cleave the His6-SUMO, FtsZ together with His-tagged SUMO protease (Ulp1) (1:100 molar ratio) was dialysed overnight at 4 °C against buffer B (50 mM Tris-HCl [pH 8.0], 300 mM KCl and 10% glycerol). To remove remaining His-tagged molecules, the sample was again passed through Ni-NTA resin, equilibrated with buffer B. The polymerization-competent fraction of the purified FtsZ was enriched by CaCl$_2$ at room temperature after buffer exchange into polymerization in buffer C (50 mM PIPES [pH 6.7] and 10 mM MgCl$_2$).

Polymerization was induced with 10 mM CaCl$_2$ and 5 mM GTP, incubated for 20 mins at RT and the polymeric fraction was collected by centrifugation at 15,000 g for 2 min and the gel-like pellet was resuspended in buffer D (50 mM Tris-HCl [pH 7.4], 50 mM KCl, 1 mM EDTA and 10% glycerol). For labelling, the thiol-reactive dye Alexa Fluor 488 C5 Maleimide (Thermo Fisher Scientific) was dissolved in dimethyl sulfoxide (DMSO) following the manufacturer's instructions. FtsZ was reduced by incubating the protein with a 100× molar excess of tris(2-carboxyethyl)phosphine (TCEP) for 20 min at room temperature. A 10× molar excess of Alexa Fluor 488 was added and extensively dialyzed against buffer D overnight at 4 °C. Remaining CaCl$_2$, GTP and free dye was removed via a PD10 desalting column and peak fraction were collected, flash frozen in liquid nitrogen and stored at −80 °C.

**Purification and fluorescence labelling of FtsAs.** FtsA was cloned into vector pMAR19, with an N-terminal TwinStrep-SUMO fusion protein plus a 5xGlycine tag for fluorescence labelling via sortagging. FtsA was expressed in *E. coli* BL21 cells, grown at 37 °C in 2× YT medium supplemented with 100 µg ml$^{-1}$ ampicillin and expression was induced at an OD600 of 0.6–0.8 with 1 mM IPTG. The protein was expressed at 18 °C and harvested by centrifugation (5000 g for 30 min at 4 °C). The pellet was resuspended in buffer A (50 mM Tris-HCl [pH 8.0], 500 mM KCl, 10 mM MgCl$_2$ and 0.5 mM DTT) supplemented with EDTA-free protease inhibitor cocktail tablets and 1 mg ml$^{-1}$ DNase I. Cells were lysed by sonication using a Q700 Sonicator equipped with a probe of 12.7 mm diameter, which was immersed into the resuspended pellet. The suspension was kept on ice during sonication (Amplitude 40, 1 s on and 5 s off for a total time of 10 min). Subsequently, cell debris was removed by centrifugation at 23,500 g for 45 min at 4 °C. The clarified lysate was incubated with IBA Lifesciences Strep-Tactin® Sepharose® resin for 1 h at 4 °C. Subsequently, the resin was washed with 40x CV buffer A and the fusion protein was eluted using buffer A containing 5 mM desthiobiotin. The protein concentration was determined with Bradford and adjusted to 12 µM with buffer A, in order to avoid precipitation of the protein. The His$_6$-SUMO protease Ulp1 was added in a 1:100 molar ratio, and the TS-SUMO tag was cleaved overnight at 4 °C, without shaking. To remove the cleaved tag and Ulp1, FtsA was subjected so size exclusion chromatography. A HiLoad 26/600 Superdex 200 Prep grade column was equilibrated with buffer B (50 mM Tris [pH 8.0], 500 mM KCl, 10 mM MgCl$_2$, 10% Glycerol and 0.5 mM DTT) and the protein was injected. The peaks containing the final protein, corresponding to monomeric FtsA, were determined via SDS-page gel-electrophoresis, pooled and concentrated as described above. For total internal reflection fluorescence microscopy FtsA was labelled with Cyanin-3 or Cyanin-5 via sortagging[39]. The 5xGly tag at the N-terminus of FtsA was conjugated to CLEPTGG-peptide, which was previously labelled via maleimide directed labelling with either sulfo-Cyanine 3-maleimide or sulfo-Cyanine 5-maleimide (Lumiprobe). Ten micrometre Sortase, 0.5 mM labelled peptide and 10 µM of FtsA were mixed together and incubated overnight at 4 °C. To remove free peptide, free dye and sortase, FtsA was subjected to another Size-exclusion on a HiLoad Superdex 200 16/600 prep grade column, pre-equilibrated with buffer B. The monomeric protein was collected, and the concentration was determined via Bradford. The concentration of dye molecules (=labelled protein) was measured by NanoDrop and the Degree of Labelling (DoL) was determined by calculating the ratio of labelled Protein:unlabelled Protein. The DoL for FtsAs was between 65–70%. To obtain the hypermorphic mutant of FtsA, R286W, pMAR19 was used as a base for site-directed mutagenesis (SDM). We replaced Arginine 286 with Tryptophan, by exchanging a single nucleotide (C→T), resulting in pMAR25. The variant of FtsA was purified in the same way as described above for the wild-type protein. To purify His-tagged variants of FtsA wt and R286W, the C-terminal amphipathic helix at position 405-420 (GSWIKRLNSWLRKEF*) of pMAR19/pMAR25 was replaced by a 6xHistidine Tag, resulting in pNB4 and pNB5 (FtsA WT-His6 and R286W-His6 respectively). The purification and labelling were performed as described above for native FtsA.

**Purification and fluorescence labelling of His-tagged SUMO-Cys (HS-Cys).** As a control for the FRET assay, we constructed a vector based on pTB146 containing only the SUMO protein, modified with an N-terminal 6xHis tag and a C-terminal Cysteine for maleimide labelling, resulting in pPR5. HS-Cys was expressed in *E. coli* BL21 cells, at 37 °C in Terrific Broth supplemented with 100 µg ml$^{-1}$ ampicillin and expression was induced at an OD600 of 0.6–0.8 with 1 mM isopropyl-β-thiogalactopyranoside (IPTG) and incubated for 3 h at 37 °C. Cells were harvested by centrifugation (5000 g for 30 min at 4 °C). Lysis and incubation with Ni-NTA-beads was performed as described before for FtsZ. The protein was eluted with buffer A (50 mM Tris-HCl [pH 7.4], 300 mM KCl and 10% glycerol) supplemented with increasing concentrations of Imidazole (50/100/150/200/250/300/400 mM). The purity of the fractions were checked by SDS Page, and the eluted fractions with 200 and 250 mM Imidazole were pooled together. Subsequently, HS-Cys was dialyzed overnight against buffer B (50 mM Tris-HCl [pH 7.4], 100 mM KCl and 10% glycerol) and finally labelled by maleimide directed labelling with sulfo-Cyanine 3 and Cyanine 5 maleimide.

Free dye and remaining traces of Imidazole were removed by Size-exclusion on a HiLoad Superdex 200 16/600 prep grade column, pre-equilibrated with buffer B. The peaks corresponding to the labelled HS-Cys were collected and concentrated with Vivaspin 20 centrifugal concentrators (5 kDa cutoff). The final concentration

and the degree of labelling were determined as described above, and the protein was stored at −80°.

**Fluorescent labelling of peptides.** The cytoplasmic peptide of FtsN with a C-terminal His6 tag and an N-terminal cysteine residue was labelled and handled as described before[4]. In short, the peptide was reconstituted to a concentration of 10 mg ml$^{-1}$ in buffer A (50 mM HEPES-KOH [pH 7.4], 150 mM KCl and 10% glycerol) and incubated with 100x molar excess of TCEP and a 10x molar excess of Cy5 overnight at 4 °C. Free dye was removed by Ni-NTA affinity (HisPur Ni-NTA Resin; Thermo Fisher Scientific) and elution with buffer A 400 mM Imidazole. Finally, the labelled peptide was dialyzed into storage buffer (20 mM HEPES-KOH [pH 6.0], 150 mM KCl and 10% glycerol) and flash frozen. The C-terminal peptide (CTP) of FtsZ conjugated to an N-terminal TAMRA dye (5-Carboxytetramethylrhodamine; TAMRA-KEPDYLDIPAFLRKQAD) was purchased from Biomatik and reconstituted in buffer A (50 mM HEPES-KOH, [pH 7.4] to a concentration of 2 mg ml$^{-1}$), flash frozen and stored at −80 °C. The sequences of all peptides used in this study can be found in Supplementary Table 3.

**Quartz crystal microbalance-Dissipation (QCM-D).** QCM-D experiments were performed with the QSense Analyzer from Biolin Scientific, equipped with silica-coated sensor (QSX 303). The sensors were cleaned for 10 s in a Zepto plasma cleaner, mounted in the QCM-D chambers and the acquisition of the experiment was first performed in the reaction buffer (50 mM Tris-HCl [pH 7.4], 150 mM KCl and 5 mM MgCl$_2$). The supported lipid membrane was formed by rupturing 0.5 mM small unilamellar vesicles (SUVs) in the presence of MgCl$_2$. The lipids used were 1,2-dioleoyl-sn-glycero-3-phosphocholine (DOPC) and 1,2-dioleoyl-sn-glycero-3-phospho-(1′-rac-glycerol) (DOPG) at a ratio of 67:33 mol%. After bilayer formations, the reaction buffer supplemented with 2 mM ATP, 2 mM GTP and 1 mM DTT was injected and the signal was recorded until a stable equilibrium was reached. Subsequently, increasing concentrations of FtsA were injected in the QCM-D chamber and changes in frequency and dissipation were monitored in real-time. The flow rate used in all experiments was 25 µl min$^{-1}$, and the temperature was set to 25 °C. To estimate the membrane-binding affinity, we extracted the frequency changes of the 5th acoustic harmonic for different FtsA concentrations and fitted a Hill equation $y = S + (E - S) \cdot (\frac{x^n}{K^n + x^n})$, where S is the starting point, E the end point, n is the Hill coefficient and k the dissociation constant. The measured binding affinity is an upper estimate, because QCM-D accounts not only on the dry molecular mass, but also on the hydration shell of the molecular assembly.

**Size exclusion chromatography with multiple angle light scattering (SEC-MALS).** 45 µg (100 µl of a 12 µM solution) of purified FtsA WT and 47 µg of FtsA R286W (100 µl of a 12.7 µM solution) was resolved on a Superdex 200 Increase 10/300 at a flow rate of 0.5 ml min$^{-1}$ at room temperature. Light scattering was recorded on a miniDAWN light scattering device (Wyatt). Changes in the refractive index were used to define the peak area, which was used to obtain the molecular mass. The analysis of the data was performed with the ASTRA software (Wyatt).

**Microscale thermophoresis (MST).** MST experiments were performed with either 50 nM FtsA WT or R286W labelled with Cyanin-5 and increasing concentrations of unlabelled FtsN$_{cyto}$. The peptide was diluted in buffer A (50 mM Tris-HCl [pH 7.4], 150 mM KCl, 5 mM MgCl$_2$ and 0.005% Tween-20). After adding the peptide to the protein, the mixtures were left to incubate for 10 min at room temperature and subsequently loaded in premium coated capillary tubes (NanoTemper). Measurements were performed with a Monolith NT.115 (Nano-Temper) equipped with a blue and a red filter set. The data was acquired with 20% MST and 20% light-emitting diode settings at 25 °C. Cy5 fluorescence was measured for 5 s before applying a thermal gradient for 30 s. Binding curves were obtained by plotting the normalized change in fluorescence intensity after 20 s against the concentration of titrated peptide. To extract the binding affinity, a Hill equation was fitted $= U + \frac{B-U}{1+(\frac{EC50}{C})^n}$, where C is the peptide concentration, U is the signal for the unbound state, B the signal for the bound state and n is the Hill coefficient. The data analysis and the fitting was performed using the MO.Affinity Analysis software.

**Cuvette FRET experiments.** For in solution FRET experiments, increasing concentrations of Cy3-FtsA and Cy5-FtsA in a 50:50 ratio were mixed in 100 µl reaction buffer (50 mM Tris-HCl [pH 7.4], 150 mM KCl and 5 mM MgCl$_2$) inside a quartz cuvette (Hellma® fluorescence cuvettes, ultra Micro). The spectrums were measured using a Spectrophotometer Spectramax M2e Plate- + Cuvette Reader (Molecular Devices). Cy3-FtsA was excited at a wavelength of 520 nm and the resulting emission spectrum was recorded from 550–700 nm in 1 nm steps. To avoid crosstalk of the excitation light, a cutoff filter was set to 550 nm. Addition of ATP or small unilamellar vesicles (SUVs) was measured individually for each concentration. Buffer controls containing the corresponding reagents and only Cy5-FtsA were measured and used as background corrections for measurements with Cy3- and Cy5-FtsA. Background corrected spectra were used to estimate

FRET efficiency by $E(\%) = \frac{Fa}{Fd+Fa} \cdot 100$, where Fa is the peak of acceptor (=Cy5) emission at 670 nm and Fd the peak of the emission of the donor (=Cy3) spectrum at 565 nm.

**Preparation of coverslips.** Glass coverslips were cleaned in piranha solution (30% $H_2O_2$ mixed with concentrated $H_2SO_4$ at a 1:3 ratio) for 60 min, and extensively washed with dd$H_2O$, followed by 10 min sonication in double-distilled $H_2O$ and further washing with dd$H_2O$. Cleaned coverslips were stored for no longer than 1 week in $H_2O$ water. Before formation of the supported lipid bilayers, the coverslips were dried with compressed air and treated for 10 min using a Zepto plasma cleaner (Diener electronics) at maximum power. As reaction chambers, 0.5 ml Eppendorf tubes without the conical end were glued on the coverslips with ultraviolet glue (Norland Optical Adhesive 63) and exposed to ultraviolet light for 10 min.

**Preparation of small unilamellar vesicles (SUVs).** For experiments without His-tagged peptides, DOPC and DOPG at a ratio of 67:33 mol% was used. To enable peptide attachment to the lipid membrane SUVs with 1 mol% dioctadecylamine (DODA)-tris-NTA (synthesized by ApexMolecular), in a ratio of 66:33:1 mol% DOPC:DOPG:Tris-NTA, were prepared. To titrate the density of Tris-NTA lipids, SUVs without and with up to 5% Tris-NTA were mixed together before supported lipid bilayer formation in the appropriate volumes. For SUV preparation, lipids in chloroform solution were added into a glass vial and dried with filtered $N_2$ to obtain a thin homogeneous lipid film. Residual chloroform was removed by further drying the lipids for 2–3 h under vacuum. Subsequently, swelling buffer (50 mM Tris-HCl [pH 7.4] and 300 mM KCl) was added to the lipid film and incubated for 30 min at room temperature to obtain a total lipid concentration of 5 mM. Tris-NTA lipids were present, 5 mM $Ni_2SO_4$ were added to the swelling buffer to load NTA groups with Nickel.

To disrupt multilamellar vesicles, the mixture was repeatedly vortexed rigorously and freeze–thawed (8x) in dry ice or liquid $N_2$. To obtain small unilamellar vesicles the liposome mixture was tip-sonicated using a Q700 Sonicator equipped with a ½ mm tip (amplitude = 1, 1 s on, 4 s off) for 25 min on ice. The vesicles were centrifuged for 5 min at 10,000 g and the supernatant was stored at 4 °C in an Argon atmosphere and used within 1 week.

**Preparation of supported lipid bilayers (SLBs).** To prepare supported lipid bilayers, the SUV suspension was diluted to a lipid concentration of 0.5 mM with swelling buffer. Vesicle rupture was induced by adding 5 mM $CaCl_2$ to the SUVs on the glass surface. The bilayers were incubated for 30 min at 37 °C, and remaining non-fused vesicles were washed away by pipetting an excess of swelling buffer (5x) on top, followed by 5x washes with reaction buffer (50 mM Tris-HCl [pH 7.4], 150 mM KCl and 5 mM $MgCl_2$) The membranes were used within 4 h after the preparation.

**TIRF microscopy.** Experiments were performed using two TIRF microscopes. The iMIC TILL Photonics was equipped with a 100× Olympus TIRF NA 1.49 differential interference contrast objective. The fluorophores were excited using laser lines at 488, 561 and 640 nm. The emitted fluorescence from the sample was filtered using an Andromeda quad-band bandpass filter (FF01-446-523-600-677). For the dual-colour experiments, an Andor TuCam beam splitter equipped with a spectral long pass of 640 nm and band pass filter combinations of 579/34 and 700/75 or 525/50 and 679/41 nm were used. Time series were recorded using iXon Ultra 897 EMCCD Andor cameras (X-8499 and X-8533) operating at a frequency of 5 Hz for standard acquisition and at 10 Hz for single-molecule tracking. The Visitron iLAS2 TIRF microscope was equipped with a 100xOlympus TIRF NA 1.46 oil objective. The fluorophores were excited using laser lines at 488, 561 and 640 nm. The emitted fluorescence from the sample was filtered using a Laser Quad Band Filter (405/488/561/640 nm). For the dual-colour experiments, a Cairn TwinCam camera splitter equipped with a spectral long pass of 565 and 635 nm and band-pass filters of 525/50, 595/50, 630/75, 670/50 and 690/50 nm was used. Time series were recorded using Photometrics Evolve 512 EMCCD ($512 \times 512$ pixels, $16 \times 16 \ \mu m^2$) operating at a frequency of 5 Hz.

**Dual-colour FtsA-FtsZ experiments.** To study colocalization and co-treadmilling of FtsA with treadmilling FtsZ filaments on supported lipid bilayers, we used Cy5-FtsA wt or Cy5-R286W (0.1–0.8 μM) and Alexa488-FtsZ (1.25 μM) in 100 μl of reaction buffer. Additionally, the reaction chamber contained 4 mM ATP and 4 mM GTP, as well as a scavenging system to minimize photobleaching effects: 30 mM d-glucose, 0.050 mg ml$^{-1}$ Glucose Oxidase, 0.016 mg ml$^{-1}$ Catalase, 1 mM DTT and 1 mM Trolox. Prior addition of all components a corresponding buffer volume was removed from the chamber to obtain a total reaction volume of 100 μl. The dynamic protein pattern was monitored by time-lapse TIRF microscopy at one frame per 2 s and 50 ms exposure time.

**FtsZ single molecule experiments.** Single-molecule experiments were performed as described previously[40]. In short, individual FtsZ proteins were imaged at single-

molecule level by adding small amounts of Cy5-labelled FtsZ (200 pM) to a chamber with 0.2 or 0.4 μM FtsA and 1.25 μM Alexa488-FtsZ.

**Single-molecule measurements of the C-terminal peptide of FtsZ (CTP).** To measure residence times of the FtsZ-CTP, we added the TAMRA-labelled CTP peptide (TAMRA-KEPDYLDIPAFLRKQAD, synthesized by Biomatik) to membranes with 1% Tris-NTA lipids. Before addition of the peptide, 1 μM of His-tagged variants of FtsA wt and R286W were added to the chambers, incubated for 20 min and washed 6x with reaction buffer. Subsequently, 1 nM of FtsZ TAMRA-CTP was added to the chamber and single molecule time lapses were acquired every 32 or 51 ms, with exposure times of 30 and 50 ms, respectively.

**Dual-colour FtsN-FtsZ and FtsN-FtsA experiments.** To study the colocalization of Cy5-labelled FtsN$_{cyto}$, we used membranes with 0.25% Tris-NTA lipids to ensure stable peptide immobilization. FtsN$_{cyto}$ at 1 μM was added to the chamber and left to incubate for 20 min to ensure homogeneous binding. Subsequently, the chamber was washed 6x with reaction buffer, to remove bulk peptide. To visualize colocalization with either FtsZ or FtsA, either a mix of FtsA and Alexa488-FtsZ or Cy3-FtsA and FtsZ was added. The concentration of FtsZ was again kept constant at 1.25 μM, whereas FtsA concentrations were titrated from 0.1–0.8 μM. The time-lapse videos were recorded for 10 min after the addition, with one frame per 2 s.

**Single-molecule experiments for confinement of FtsN$_{cyto}$.** To study the interaction of single molecules of Cy5-labelled FtsN$_{cyto}$, we also used membranes with 0.25% Tris-NTA lipids. This time 1 μM of unlabelled FtsN$_{cyto}$ supplemented with 50 pM of Cy5-labelled FtsN$_{cyto}$ were added to the chamber and incubated for 20 min, followed by 6x washes with reaction buffer. Subsequently, 0.2 μM of either FtsA WT or FtsA R286W and 1.25 μM Alexa488-FtsZ were added to the chamber and pattern formation was recorded for 10 min. Single molecule time lapses were acquired every 32 or 51 ms, with exposure times of 30 and 50 ms, respectively.

**Single-molecule experiments on FtsA WT and FtsA R286W.** To study the behaviour of single molecules of FtsA WT and FtsA R286W, 0.1 μM of the unlabelled protein supplemented with 35 pM Cy5 of the respective FtsA variant were added to the reaction chamber. After 5 min of incubation, single molecule time lapses were acquired every 125, 250, 500, and 1000 and 2000 ms, with an exposure time of 50 ms. Subsequently, the bulk concentration of FtsA was increased to 0.2/0.4/0.8 μM and time lapses were repeated as described above.

**FRAP and FRET experiments on SLBs.** To measure the membrane residence time and self-interaction of FtsA WT and FtsA R286W, acceptor (Cy5) photobleaching experiments were performed. Equimolar concentrations of Cy3- and Cy5-labelled FtsA, supplemented with 20% unlabelled FtsA were used to study FRET and FRAP. Five pre-bleach frames were acquired, followed by acceptor photobleaching of a rectangular ROI with 40% 641 nm laser power and a dwell size of 1 μs/pixel, 75% overlapping lines. The recovery of the signal or the increase in donor intensity were measured with either two frames or one frame per second. The different acquisition rates were implemented, due to the accelerated recovery of R286W compared to wt. To measure effects of FtsZ, 1.25 μM of unlabelled FtsZ was added to the membrane and FRET/FRAP was measured again. To quantify effects of FtsN$_{cyto}$, SLBs with 0.25% Tris-NTA lipids were pre-equilibrated with FtsN$_{cyto}$ before additions of FtsAs. Subsequently, 1.25 μM FtsZ was added as well to quantify effects of the combined presence of FtsN$_{cyto}$ and FtsZ.

**SLB experiments of His-tagged FtsAs.** To study colocalization of His-tagged variants of FtsA, 0.5–1 μM of Cy5 labelled His-tagged FtsAs were added to the chamber, incubated for 20 min and washed 6x with reaction buffer. Subsequently, 1.25 μM Alexa488-FtsZ was added to the chamber and pattern formation was recorded for 20 min. To control the density of membrane-bound FtsA, SUVs without and with Tris-NTA lipids were mixed to obtain the respective Tris-NTA concentrations. To perform FRET/FRAP experiments, equimolar concentrations of Cy3 and Cy5 labelled His-tagged FtsAs (total 0.5–1 μM) or His-SUMO-Cys were added to a chamber and treated as above. To study effects of FtsZ, 1.25 μM unlabelled FtsZ were added to the chamber and recorded for 20 min. FRAP experiments were performed as described before.

For measurement with both, FtsA-His6 and FtsN$_{cyto}$-His, we used SLBs containing 1.5% Tris-NTA membranes and incubated them for 20 min with 1 μM FtsA-His6 (0.5 μM of each Cy3- and Cy5 His-FtsA) and 1 μM FtsN$_{cyto}$. If all components bind to Tris-NTA lipids with the same affinity, this should lead to a membrane covered by equimolar amounts of FtsA-His6 and FtsN$_{cyto}$. Subsequently, the membranes were washed 6x with 200 μL and pattern formation was triggered by addition of 1.25 μM FtsZ (or Alexa488-FtsZ).

**Image processing and analysis.** For data analysis, the movies were imported to the FIJI software[41]. For data analysis, raw, unprocessed time-lapse videos were used. All micrographs in the manuscript were processed with the walking average plugin of ImageJ, averaging the signal of four consecutive frames, and contrast was optimized for best quality.

**Colocalization analysis**. Time-lapse videos were first intensity-corrected and contrast-enhanced to avoid bleaching effects and simplify subsequent analysis. To remove contributions of X–Y drift, the videos were processed with the Linear Stack Alignment with SIFT plugin. Proper alignment was checked with the 3TP align plugin (J. A. Parker; Beth Israel Deaconess Medical Center, Boston). Subsequently, regions of interest (ROIs) in the center of the stacks were chosen for colocalization analysis. The Pearson's correlation coefficient (PCC) was quantified with the Image CorrelationJ 1o plugin. To extract information about the relative ratio of FtsZ/FtsA molecules (slope of linear regression), we also used the Image CorrelationJ 1o plugin. As an output, the plugin provides a scatterplot of FtsZ vs FtsA intensities, to which we fitted a linear slope $y = k \cdot x + d$, where $k$ is the slope and $d$ the offset. The slope was used as an estimate for the ratio of FtsA molecules below FtsZ filaments[42].

**FtsN recruitment rate quantification**. To estimate the rate of $FtsN_{cyto}$ recruitment towards FtsA/Z co-filaments, we measured the PCC after adding FtsA/Z to a membrane homogeneously covered with $FtsN_{cyto}$ and fitted a power law equation $y = a \cdot (1 - e^{-b \cdot t}) + c$, where a is the starting point, b is the rate and c is the offset, to the increasing PCC values after protein addition and extracted the recruitment rate.

**Treadmilling and temporal PCC analysis**. Treadmilling dynamics, were quantified using an automated image analysis protocol previously developed by our group[24]. To visualize colocalization of the co-treadmilling FtsZ & FtsA filaments, we used dual-colour videos obtained at an acquisition rate of one frame per 2 s. The two channels were aligned using FIJI's 3TP align plugin. Both channels were then subjected to the image subtraction protocol and colocalization was measured as described above.

**FtsZ autocorrelation analysis**. To measure reorganization dynamics of FtsZ filaments, we used a temporal correlation analysis based on the Image CorrelationJ 1o plugin. We quantified the PCC between the first frame to subsequent frames with increasing time lag ($\Delta t$). The decrease in the PCC was plotted against $\Delta t$ to obtain autocorrelation curves. Slower decay indicates more persistent structures. The rates of decay were extracted by fitting monoexponential decay to the autocorrelation curves $y = a \cdot e^{(-b \cdot t)} + k$, where a is the starting point, b is the decay rate and k is the final offset. The half time of the monoexponential decay was calculated via the decay rate.

**Transient confinement analysis of $FtsN_{cyto}$**. Single-molecule experiments with $FtsN_{cyto}$ were tracked using the TrackMate plugin from ImageJ[43]. Non-moving particles and short tracks (below 1 s) were filtered out and the data exported as.xml files. To analyse transient confinement periods of $FtsN_{cyto}$ to FtsZ/FtsA cofilaments we used the packing coefficient ($p$) to identify when diffusing $FtsN_{cyto}$ molecules switch between free diffusion and confined motion. The packing coefficient is defined as the length of the trajectory in a short time window and the surface area that it occupies. This gives an estimate of the degree of free movement that a molecule displays in a period independently of its global diffusivity. This approach is adapted from Renner et al. and implemented here as an easy-to-use Python script[44]. The packing coefficient is computed for each time point as:

$$p = \sum_{i}^{i+n-1} \frac{(x_{i+1} - x_i)^2 - (y_{i+1} - y_i)^2}{S_i^2} \quad (1)$$

Where $x_i, y_i$ are the coordinates at time $i$, $x_{i+1}$, $y_{i+1}$ are the coordinates at time $i+1$, n is the length of the time window, and $S_i$ is the surface area of the convex hull of the trajectory segment between time points $i$ and $i+n$. Periods of confinement are identified by setting a threshold corresponding to a certain confinement area size, since $p$ scales with the size of the confinement area. Then it is possible to calculate the frequency and duration of confinement periods and to localize them in space. Each position will have a characteristic $p$, considering the behaviour of the following $n$ positions. This approach overcomes the limitations of using MSD calculation, which overlooks transient confinement periods. Nevertheless, the Brownian diffusion trajectories can temporarily mimic confinement due to random fluctuations of the length of the displacements. However, the amplitudes and durations of these fluctuations are most of the time smaller and shorter than the ones associated with real non-Brownian transient motion. Therefore, the use of a threshold value of $p$ ($p_{thresh}$) and a minimal duration above this threshold ($t_{thresh}$) can suppress the detection of apparent non-random behaviours without excluding the detection of real confinement. These parameters depend on the acquisition frequency (which will affect the length of the time window) and the characteristic time of confinement. Too large windows will not properly detect the confinement period, while the statistical uncertainty increases in shorter windows. Thus, to accurately detect confinement periods, the window size should be adjusted accordingly to the acquisition rate. To detect periods of confinement, we set $p_{thres}$ to 1000, which corresponds to confinement areas of roughly < 50 nm, and a $t_{thres}$ of 0.25 s, which corresponds to five and eight frames when using 51 ms and 32 ms acquisition rates, respectively. The thresholds for confinement and time were chosen after manual inspection of tracks and corresponding confinement events.

Finally, mean confinement times were extracted by fitting a monoexponential decay function to histograms of confinement times of individual experiments. To validate the performance of our code, we simulated single-molecule tracks that switch between free diffusion and transient confinement periods using FluoSim[45]. To simulate appropriate tracks some parameters were kept constant for all tracks: 50 molecules/FOV; a $D_{coeff}$ out- and inside 0.2 $\mu m^2/s$, and the crossing probability was set to 1. To test the performance of our code, we varied binding rates from 0.1–0.5 $s^{-1}$, unbinding rates from 0.5–3 $s^{-1}$ and the trapped $D_{coeff}$ from 0.002–0.008 $\mu m^2/s$. The values were chosen according to previously acquired and published data[4]. At low diffusion coefficient for trapped molecules (< 0.005 $\mu m^2/s$) confinement periods were identified with a marginal error of ±0.04 s, whereas the performance of the routine suffered slightly when increasing the diffusion coefficient of trapped molecules (> 0.008 $\mu m^2/s$), but still resulted in values close to the ground truth (±0.1 s). We also used these simulated tracks to fine-tune the thresholds for $FtsN_{cyto}$ confinement time analysis. The source code can be found in https://doi.org/10.5281/zenodo.6397261[46].

**Single-molecule analysis of FtsZ and FtsA**. Single molecules of FtsA or FtsZ were tracked using the TrackMate plugin from ImageJ[43]. To obtain the residence time of FtsZ and FtsA, we performed a residence time analysis as described before[40,47]. Shortly, single molecules were imaged at different acquisition rates (0.1–2 s) and the lifetime of the molecules was extracted from each data set. To account for photobleaching effect, the obtained lifetimes were plotted against the acquisition rate. Then, we fitted a linear regression to this data and the photobleach corrected lifetime was calculated by taking the inverse of the slope of the linear regression. Furthermore, we extracted the diffusion coefficient of FtsA single molecules at increasing concentrations. For this, we filtered the obtained data, by considering only trajectories which are present on the membrane for more than 0.4 s. This low filter threshold was necessary, due to the very short lifetime of FtsA R286W molecules at low concentrations. Subsequently, the diffusion coefficient of FtsA molecules was estimated by fitting an MSD curve to each individual trajectory.

**Quantification of the FRET efficiency, diffusion coefficient ($D_{coeff}$) and the off-binding rates ($k_{off}$) from FRAP experiments**. To estimate the degree of self-interaction of FtsA WT and FtsA R286W, we used a photobleaching approach as outlined above. Bleaching the acceptor dye leads to an increase in the donor intensity, which can be used to quantify the Foerster Resonance energy transfer (FRET)[48]. FRET efficiency was quantified with $E[\%] = \frac{Ipost}{Ipost + Ipre} \cdot 100$, where $I_{post}$ is the intensity of the acceptor (Cy3) after bleaching of the donor and $I_{pre}$ is the acceptor intensity before bleaching the donor. While quantifying membrane binding dynamics of FtsA, we realized soon, that the recovery of FtsA was achieved by two different mechanisms: simple on- and off-binding to the membrane and lateral diffusion of the protein along the SLB. To extract the contribution of both processes, we adjusted a routine recently published by Gerganova et al.[33]. In short, the code provides the contribution of both modes of recovery by analysing the shape of the fluorescence recovery profile. For simple on/off binding, the profile shape during recovery does not change (compare Fig. S4h left). Contribution of diffusion leads to a change of the slope of the outer borders of the bleached region (S4h right). Thus, by measuring the change of the slope of the border recovery, diffusion and simple on/off binding can be distinguished. We developed "FRAP-diff", a graphical user interface, which uses the original fit routine by David Rutkowski. It can be used directly on.tiff movies with annotated bleach regions as ImageJ ROIs. In addition, we added optional bleach correction, user selectable projection axis (x or y), and optional mirroring of the recovery profiles, if bleaching was not symmetric. The original fit routine for estimating the recovery and diffusion dynamics can be found at https://github.com/davidmrutkowski/1DReflectingDiffusion. FRAPdiff is available at https://doi.org/10.5281/zenodo.6400639[49] under the GPLv3 open-source license.

**Calculation of spacing and theoretical maximum packing of membranes with His-FtsAs**. To calculate the theoretical spacing of His-tagged FtsAs or the His-SUMO control, we consulted a previous QCM-D study using Tris-NTA lipids and a His-tagged version of ZipA[50]. The spacing in $nm^2$ was calculated by $y = \frac{2}{\sqrt{3}} \cdot \frac{1}{6.022 \cdot x \cdot 0.001}$, where x is the protein density in pMol $cm^{-2}$ which can be estimated from the Tris-NTA lipid density[51].

To estimate the theoretical maximum packing of membranes containing Tris-NTA lipids, we can assume the area of a single FtsA molecule to be ~30 $nm^2$ and the area of a single lipid molecule to be 0.5 $nm^2$. Accordingly, a single FtsA would occupy 60 lipid molecules in the membrane and we should reach the theoretical maximum coverage of the membrane with His-FtsAs at a Tris-NTA density of 1.66%, which corresponds to the Tris-NTA density at which we see a saturation of the measured FtsA intensity (Fig. S4b) as well as the FRET signal (Fig. S4k). For the FRET control experiments with His-SUMO-Cys, we estimate a four times smaller area for His6-SUMO Cys (~7 $nm^2$) and thus increase the maximum Tris-NTA density to 5%. Only at this density, we observe a small increase in FRET (Fig. S4k).

**Statistics and reproducibility**. Statistical details of the experiments are reported in the figure captions and the corresponding Source Data Files. Reported $P$-values were calculated using a two-tailed Student's $t$-test for parametric distributions. Sample sizes are at least two independent experiments. No statistical test was used to determine sample sizes. Biological replicates are defined as the number of independent experiments in which a new reaction chamber was used. Independent experiments in some cases were performed on the same coverslip, which could fit up to three reaction chambers. Unless otherwise stated in the figure captions, the graphs show means ± standard deviation and the error bars were calculated and are shown based on the number of independent experiments, as indicated. The distribution was assumed to be normal for all biological replicates.

**Reporting summary**. Further information on research design is available in the Nature Research Reporting Summary linked to this article.

## Data availability

Source data for Figs. 1–4 and Supplementary Figs. 1–5 are provided with the paper. Raw microscopy images as well as the Jupyter notebooks used for plotting and fitting can be found at https://doi.org/10.15479/AT:ISTA:10934.

## Code availability

The code used to measure confinement is available online at https://doi.org/10.5281/zenodo.6397261[46]. Our customized code to analyse contributions of off binding rates and diffusion coefficients for bleaching recovery (FRAPdiff) can be found at https://doi.org/10.5281/zenodo.6400639[49]. The treadmilling analysis code is available online at https://doi.org/10.5281/zenodo.6397778[24]. The updated and corrected code to quantify single molecule lifetimes has been uploaded and is available at https://github.com/paulocaldas/PhotobleachingCorrectionSPT.

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

## Acknowledgements
We acknowledge members of the Loose laboratory at IST Austria for helpful discussions—in particular L. Lindorfer for his assistance with cloning and purifications. We thank J. Löwe and T. Nierhaus (MRC-LMB Cambridge, UK) for sharing unpublished work and helpful discussions, as well as D. Vavylonis and D. Rutkowski (Lehigh University, Bethlehem, PA, USA) and S. Martin (University of Lausanne, Switzerland) for sharing their code for FRAP analysis. We are also thankful for the support by the Scientific Service Units (SSU) of IST Austria through resources provided by the Imaging and Optics Facility (IOF) and the Lab Support Facility (LSF). This work was supported by the European Research Council through grant ERC 2015-StG-679239 and by the Austrian Science Fund (FWF) StandAlone P34607 to M.L. and HFSP LT 000824/2016-L4 to N.B. For the purpose of open access, we have applied a CC BY public copyright licence to any Author Accepted Manuscript version arising from this submission.

## Author contributions
P.R., N.B. and D.M. performed the experiments. P.R. analysed the data. P.C. and C.S developed codes for data analysis. M.L., P.R., and N.B. wrote the manuscript. All authors discussed the results. M.L., N.B. and P.R. developed the project. M.L. and N.B. conceived and supervised the study.

## Competing interests
The authors declare no competing interests.
