## [Peer Review File · Nature Communications]

REVIEWER COMMENTS

Reviewer #1 (Remarks to the Author):

The authors use Fluorescence Microscopy and QCM-D in order to study some important aspects of the bacterial division mechanism. I would like to make the following comments:

Major

- in the Abstract, the authors should rephrase the statement "By comparing..." in Lines 21-24, in order to make it more clear
- in Results, Line 88, the expression "two different time scales" is misleading/needs clarification
- Line 105: "...remained unchanged"; by inspecting Fig.1c and comparing it to Fig.1g this conclusion is not as self-evident as the authors claim it to be. Given the X-axis scale is in μm , and comparing it to protein dimensions, something more needs to be said at this point
- also, Lines 166-169 & 365-366, by inspection, rings for example, are visible in both Fig.1c and Fig.1g
- Line 127: "...decreased self-interaction...", does it mean that the mutant FtsA molecules are monomers in bulk (so are the wt ones, Line 262) or something else ?
- Lines 157-165: how was the binding curve in Fig.1k obtained? are sequential additions of higher and higher concentration of FtsA made on the same SLB (as in Fig.S1h) or new surfaces were exposed to each different concentration? in principle, the two Figures are not comparable. (also, is there a buffer "wash" step in the acoustic experiments of Fig.S1h?)
- Line 790: the formation of the SLB should be proven by the corresponding QCM experiment (shown as SI maybe)
- is there an explanation why the, assumed, formation of oligomers by the wt FtsA stably forms (does not all wash away) on the SLB and not by the mutant? is the presence of DOPG necessary for these interactions and why ?
- what is the estimated ratio of lipids-per-protein on the SLBs ? it might be of help to estimate (from geometrical considerations) whether, using say 1-2% of NTA lipids or more , is meaningful

Minor

- Fig1i., needs correction in Y-axis
- at which acoustic harmonic were the shown data recorded; please add
- I think that more detailed/clearer cartoon(s) of the "geometry" of the experiments (showing the surface+proteins) would help to make the manuscript descriptions far more readable without forcing someone to look for other resources. The Mw, dimensions and pI of the proteins would be useful information too – they are vital for supporting/explaining some of the results .

In conclusion, this study contains a lot of new work in an area where it is much needed; I read the manuscript with great interest. The authors tried to build their case slowly putting into it a lot of experimental evidence. The fluorescence technique was indeed greatly exploited but the QCM was not; much information is left untouched. Nevertheless, many findings are very interesting and entirely novel and they would make a valuable contribution to the field. In my opinion, the manuscript can be accepted for publication after consideration is given to the above-mentioned Comments._

Reviewer #2 (Remarks to the Author):

Comments on NCOMMUN 341989

This paper investigates the interaction of FtsA and its variant R268W with lipids, FtsZ and FtsN on a lipid surface. The variant is known to bypass other division proteins and to cause premature division and to favor the monomeric form of FtsA due to its reduced ability to polymerize. FtsA binds to membranes in the absence of FtsZ and assembles with FtsZ in its presence. R268W migrates together with FtsZ at a range of concentrations whereas the wt distributes on the membranes at higher concentrations. The affinity of WT and variant for membranes is similar as was already shown by Krupka et al., 2017. Affinities for the C-terminal peptide of FtsZ seemed to be similar for WT and variant. The hypothesis is that WT FtsA forms minirings on the lipid layer that limits the number of associated FtsZ filaments, whereas R268W assembles in aligned structures that allow more FtsZ filaments to bind at the higher FtsA proteins concentrations (based on EM images from Krupka et al 2017). R268W recruited the cytoplasmic domain of FtsN faster than wt FtsA without having a higher affinity for FtsN. The new information of this paper is that R268W has a much shorter residence time and diffuses much faster on the lipid bilayers than the WT and that WT and variant have the same affinity for the C-term of FtsZ and for the cytoplasmic domain of FtsN. Based on the EM images of Krupka, the N-term of FtsA should be inside the FtsA ringlets and CY3 and Cy5 could be very close together (3.5 nm) from which a very high FRET efficiency is to be expected. However, this is diluted since maximally 50% of the molecules contribute to the FRET signal. At high FtsA concentrations molecules from different ringlets can also contribute to the FRET explaining the increase in FRET efficiency as is observed. The R268W mutant has a much lower FRET signal, which might be explained by its short residence time on the membranes. When the protein is permanently attached to the membrane by a His-tagged lipid it still has a very low FRET efficiency (about 10%). Based on the hypothesis that it forms arcs and bundles and collects FtsZ that also forms bundles and the drawing in the model that predicts that every R268W protein is bound to a FtsZ molecule, I would expect the FRET efficiency to be higher. I have the impression that R268W is not associating with itself and that FtsZ has no effect on the association. In that sense it is a bit strange to depict in the model a row of R268W molecules that clearly interact. As soon as the cytoplasmic domain of FtsN is added WT and R268W become very similar with respect to FRET efficiency and diffusion rate. Only the off rate of R286W is still higher if it is not forced to bind to the membrane. Binding of FtsN apparently reorientates the R268W molecules and presumably also the WT molecules. As mentioned in the discussion, it has recently been found that FtsN stimulates the conversion of FtsA ringlets to double filaments.

It is a pity that the authors did not add FtsN to the membrane immobilized system as well.

Overall, I think that the interpretation of the data is to a large extent correct and fit the model (maybe some adjustments can be made based on the thoughts above and below).

Minor remarks:

In Fig. 1c at high FtsA concentrations one can see that the fluorescence is completely distributed over the membrane apart from some bright spots. This suggests that FtsA cannot oligomerize under these conditions. Perhaps it is good to mention explicitly that the minirings of FtsA are less than 20 nm in diameter and therefore not visible on the lipid bilayer?

Also, the FtsZ filaments look different in the image at higher FtsA concentration, fewer ringlets. In figure S1 you show that the dynamics of FtsZ do not change. Does this mean that the difference in morphology is just coincidentally? In Fig 2b I see the same, so no coincidence?

Figure S1i Y-ax should be autocorrelation.

Line 142: maybe better? In particular, previous yeast two-hybrid experiments suggested that FtsA R286W has a higher affinity for FtsZ than the wild-type protein.

Line 150: The difference in residence time of the FtsZ peptide on FtsA WT or mutant is not statistically significant. This is caused by the large SD of the WT. If the WT would have the same SD as the mutant, the difference in dwelling time would be about 18%. Would that be sufficient to explain a higher affinity or is it something that can be ignored? FtsZ would never be a monomer when it makes a functional interaction with FtsA. As a polymer the FtsZ concentration would be locally very high, then a small increase in affinity might be meaningful?

Fig 2c and d. The X-axis title of c is overlapped a bit by d.

Line 249: the number of digits indicates precision; therefore, it seems logical to write that FtsA retention time at 0.8 μM is 9.4 ± 0.3 s. Otherwise, you suggest that the precision is better at higher FtsA concentrations for which I do not immediately see a reason...

Line 248: Unfortunately, I do not have access to your paper "Single-molecule measurements to study polymerization dynamics of FtsZ-FtsA copolymers." which should describe how you measure residence time. Consequently, I do not immediately understand that you measure absence of diffusion for FtsA WT $0.004 \mu\text{m}^2/\text{s}$ but a residence time of 10 sec. based on your movie I would define residence time. The time FtsA is bound to the lipid layer before it dissociates into the medium. If I am right, it might be helpful for the reader to add such a definition to the paper?

"

Line 295: Förster. Keep pressing your finger on the o key and a menu will pop up that allows you to choose different o's.

Line 260: either Cy-5 (acceptor) or Cy3 (donor). Otherwise, the reader must search for this information, which makes it difficult to understand fig 3 d and so on.

Fig. S3C, the legend and word FRET-threshold are for some reason not completely visible.

**In previous experiments it was shown that the FtsA variants have the same affinity for the membrane. In the cuvette experiment where you add SUVs and FtsA variants and assuming that the labeling efficiency of both proteins was the same, you would expect the same amount of FRET if these proteins would be able to interact with each other in the same manner, is it not? According to the literature FtsA forms minirings and R286W forms tightly packed filaments and arcs. Yet the WT achieves a higher FRET efficiency. Initially I found that counterintuitive. But since you measure the time average FRET, the lower retention time of R286W probably explains the lower FRET despite the higher change of Cy3 and 5 to see each other in packed filaments than in ringlets. In the manuscript, you assume that FtsA behaves as described by Krupka in 2017 using EM. But you have the proteins labeled. How do you know that this is not affecting the assembled conformation of the proteins? The difference in FRET suggest that these variants interact differently (as expected). Have you thought of trying alphaFold (-
https://colab.research.google.com/github/sokrypton/ColabFold/blob/main/AlphaFold2_c/complexes.ipynb#scrollTo=g-rPnOXdjf18) with the variants to see whether any structural differences might be able to explain your results?**

Interestingly, the FRET signal is almost the same for WT and R286W in the presence of FtsZ and FtsN as if binding of FtsN forces the two proteins in a similar orientation/interaction. Also, the diffusion coefficients are more similar. You could try to look at the combination of the cytoplasmic domain of FtsN and the FtsA variants in alpha-fold as well.

In the model that explains your observations, you assume that FtsA form ringlets.

According to the paper of Krupka these are 20 nm 5 FtsZ molecules span 20 nm, so in the drawing you have every second FtsZ molecule binds an FtsA, but it should be every 5th FtsZ molecule binds an FtsA.

Line 873: FtsZ was added instead of were added?

Reviewer #3 (Remarks to the Author):

The authors present an interesting study of the *in vitro* assembly dynamics of *E. coli* FtsA, focussing on the mechanism of assembly and co-treadmilling with FtsZ. In particular they investigate how these assembly dynamics are modulated by the FtsA R286W gain of function mutant, which can support division without *E. coli*'s other membrane anchor, ZipA, and by interactions with the division-activating protein FtsN. The authors principal findings are separated into those concerning the function of the FtsA R286W mutant, and those concerning FtsN activation of FtsA.

R286W

They find that FtsA R286W colocalizes and co-treadmills much more precisely with FtsZ over a wider range of FtsA concentrations. Fitting with this they find that FtsA R286W also turns over much quicker from the membrane, and diffuses much faster. Altogether this supports a model where inhibition of more stable large arcs/ circles of FtsA common in the WT is inhibited in this mutant, and this allows FtsA R286W to track FtsZ localization more accurately, likely explaining why it acts as a gain of function mutant. A preprint from the Lowe lab supports the underlying structural hypothesis; this study provides excellent complementary analysis linking the new structural information to the resulting protein localization dynamics which are known to be central to the function of both FtsA and FtsZ.

FtsN

The FtsN results I found a little trickier to follow. Partly, I think this is because the implications of the results are a bit more complex. Interestingly the authors find that FtsN probably does lead to disassembly of FtsA oligomers, as previously speculated – rather, self interaction, measured via FRET, actually increases. However, the increased self interaction actually leads to faster FtsA turnover and diffusion on the membrane (Figure 3), which naively is probably the opposite of what one would predict - larger assemblies should be more stable – but again fits well with the recent Lowe lab work, which shows addition of FtsN to FtsA forms assembly of actin like double filaments instead of mini-rings. Excitingly, the Radler work suggests then that the FtsA+FtsN double filaments are likely much more dynamic than the FtsA arcs, allowing more accurate co-treadmilling/ co-assembly with FtsZ at the septum – as summarized in their cartoon.

Overall I really enjoyed reading this work and I recommend accepting it once the below minor criticisms are addressed. I have essentially no technical criticisms of the manuscript of substantial consequence – I was impressed with the level of experimental care and detail, and the overall clarity of the figures.

Comments to be addressed:

Please make clear that the work has been done in *E. coli*. In the title – please change to “*In vitro* reconstitution of *Escherichia coli* divisome activation” or similar. Also clarify in the abstract. In the introduction, eg division is introduced as “Divisome assembly is initiated by the simultaneous accumulation of FtsZ, FtsA and ZipA at midcell, where they organize into the Z-ring, a composite cytoskeletal structure of treadmilling filaments at the inner face of

the cytoplasmic membrane (Fig. 1a)." with no mention of the organism. In fact the organism is only mentioned once rather late (line 56) in the intro. The *ftsA* R286W gain of function mutant has, to the best of my knowledge, only been characterized in *E. coli*, FtsN is limited to the Proteobacteria, and ZipA is primarily characterized only in the context of *E. coli*. Therefore it is not justified to draw inference beyond this organism - although I would certainly be interested to hear the authors speculation on implications for organisms which lack FtsN. This over-generalism is quite annoying to the non-*E.-coli* biologist! Please address this.

Implications of the study for activation of FtsA. I struggled a bit with understanding how the FtsA R286W findings relate to the FtsA WT + FtsN experiments and specifically to the mechanisms of FtsA activation. The authors mention early "By comparing the properties of wildtype FtsA and FtsA R286W, a hyperactive mutant that represents the activated state of the protein" (line 75) – I don't think this is accurate, and also I think their results argue against the simplest version of this statement in some quite interesting ways . FtsA R286W has been characterized as a gain of function mutant which can support division in the absence of ZipA. It has previously been speculated that this may be because FtsA R286W is the activated form of FtsA. Actually the recent Lowe preprint seems to argue against this. If the active form of FtsA is double filaments, and the inactive form is arcs, FtsA R286W without FtsN definitely does not look like the active form, instead it looks like more compact arcs (Figure 2), which transition to the active form more easily on addition of FtsN. Furthermore the authors results also seem to argue against this simplest interpretation. FtsA R286W both turns over on the membrane and diffuses incredibly fast compared to WT FtsA (Fig 2b). If FtsA R286W is the active state of FtsA, and FtsN indeed promotes activation of FtsA, then one would expect the dynamics of FtsN to FtsA WT to approach the fast turnover of FtsA R286W – actually addition of FtsN to FtsA appears to cause the turnover and diffusion of FtsA to slow down a little bit (Fig 2i,j). This seems to be the opposite to the "FtsA R286W is just FtsA's active state" model. Certainly the authors should clarify what they mean on line 75, but I would also suggest that the authors consider adding some discussion of the implications of their study for the relationship between FtsA R286W and the FtsA WT active state. Also to discuss the lack of similarity between the FtsA + FtsN and FtsAR286W results – it may be that their interpretation of these data are very different to mine but I would like to know how they do interpret these very interesting results

Related point: I found it hard to make key comparisons of results with FtsA to results with FtsA R286W in the left and right panels of Figure 3hij. I'm not sure what the best solution is but maybe adding gridlines to the graphs would help compare between them? Are there any summary statistics that can be extracted from those concentration dependent data to aid comparison between the 8 different conditions. Please consider using more distinct colour maps in Fig 3hij, the different shades of blue/ purple, and different shades of green or red I struggle to tell the difference between, which made it quite tricky to interpret those graphs.

We wish to thank all reviewers for their service and their encouraging feedback. Please find a point-by-point response – in blue font - to their comments below.

Reviewer #1 (Remarks to the Author):

The authors use Fluorescence Microscopy and QCM-D in order to study some important aspects of the bacterial division mechanism. I would like to make the following comments:

Major

- In the Abstract, the authors should rephrase the statement “By comparing...” in Lines 21-24, in order to make it more clear

We rephrased the sentence.

- In Results, Line 88, the expression “two different time scales” is misleading/needs clarification

We removed the expression to avoid confusion.

- Line 105: “...remained unchanged”; by inspecting Fig.1c and comparing it to Fig.1g this conclusion is not as self-evident as the authors claim it to be. Given the X-axis scale is in μm , and comparing it to protein dimensions, something more needs to be said at this point

We agree with the reviewer and rephrased the sentence to describe our observations more accurately.

- also, Lines 166-169 & 365-366, by inspection, rings for example, are visible in both Fig.1c and Fig.1g

Here, we are referring to FtsA minirings of only about 20 nm that were previously observed by electron microscopy (see references 12 and 36). Importantly, these structures are too small to be resolved by fluorescence microscopy and are qualitatively different to the FtsZ-dependent large-scale assemblies (rings with a diameter of around 1-1.5 μm) we observe in our experiments. We apologize if this was not clear in the previous version of manuscript and modified the accordingly (see lines 171-175).

- Line 127: “...decreased self-interaction...”, does it mean that the mutant FtsA molecules are monomers in bulk (so are the wt ones, Line 262) or something else?

*Earlier studies found that the FtsA R286W variant (among others) exhibits weaker self-interaction, which correlates with its ability to bypass other cell division proteins (also see lines 59-65 of our updated manuscript). Therefore, we expected to observe weaker self-interaction than for FtsA WT. In fact, we found that **both** FtsAs are monomeric in solution, as demonstrated in our SEC-MALS experiments with FtsA WT and R286W (Figure S3a), while they show different degree of oligomerization on membranes (Fig. 3).*

- Lines 157-165: how was the binding curve in Fig.1k obtained? are sequential additions of higher and higher concentration of FtsA made on the same SLB (as in Fig.S1h) or new surfaces were exposed to each different concentration? in principle, the two Figures are not comparable. (also, is there a buffer “wash” step in the acoustic experiments of Fig.S1h?)

Fig. 1k shows the frequency changes from two independent experiments for the two versions of FtsA. Figure S1j shows exemplary measurements from QCM-D experiments for WT and R286W. Here, in each experiment an SLB on the QCM-D sensor was formed, then washed with buffer

containing nucleotides. After bilayer formation, the membrane was exposed to increasing concentrations of FtsA until equilibrium was reached, and changes in frequency/dissipation were measured (see updated Figure S1j).

As we could not see any additional binding at concentrations higher than $0.8 \mu\text{M}$ FtsA, we decided to use this concentration as maximum in our experiments.

- Line 790: the formation of the SLB should be proven by the corresponding QCM experiment (shown as SI maybe)

Fig S1j now contains also the initial stages of the QCM-D experiment, showing formation of the lipid bilayer with a characteristic shift in frequency (-30 Hz) and dissipation (0.3 ppm).

- is there an explanation why the, assumed, formation of oligomers by the wt FtsA stably forms (does not all wash away) on the SLB and not by the mutant? is the presence of DOPG necessary for these interactions and why ?

The observation that WT FtsA is not washed out as efficiently as the mutant protein is consistent with our observation of a much-reduced turnover of the wildtype protein. We see much slower desorption in WT FtsA case, which can be attributed to its intrinsic longer life-time on membranes (consistent with our FRAP and single-molecule experiments) or the formation of large-scale assemblies on membrane surface.

The lipid composition used in our experiments is motivated by a previous paper (Vecchiarelli et al. Molecular Microbiology 2014) that identified it as a mimic for the more complex mixture found in the E. coli membrane. As FtsA binds to the membrane surface via a C-terminal amphipathic helix, the negative charge provided by DOPG provides stable attachment of FtsA to the membrane surface.

- What is the estimated ratio of lipids-per-protein on the SLBs? it might be of help to estimate (from geometrical considerations) whether, using say 1-2% of NTA lipids or more, is meaningful

Assuming an FtsA monomer to have a roughly ellipsoid shape (width = 5 nm , length = 7.8 nm), it covers about 30 nm^2 of the membrane surface. An area of $1 \mu\text{m}^2$ could therefore occupy maximally $3.3 \cdot 10^4$ FtsA molecules in case of a hexagonal packing, which we can assume for FtsA R286W. At the same time, $1 \mu\text{m}^2$ contains around $2 \cdot 10^6$ lipid molecules, assuming an area of $0.5 \text{ nm}^2/\text{lipid}$. Accordingly, we expect a theoretical maximum coverage of the membrane at 1.66% Tris-NTA lipids and a corresponding lipids-per-protein ratio of about 1:60.

Figure 1: Distance measurements of single FtsA molecule via Chimera.

This theoretical value corresponds to our experimental observations, as we see a saturation of fluorescence intensities at Tris-NTA densities higher than 1.5% (Fig. S4b) and a plateau of the FRET efficiencies at 1.5-2.0% Tris NTA (Fig. S4k).

We thank the reviewer for this insightful comment and now include this point in the corresponding methods section (lines 1361-1374).

Minor

- Fig1i., needs correction in Y-axis

We corrected the typo.

- at which acoustic harmonic were the shown data recorded; please add

We used the 5th acoustic harmonic to quantify the QCM-D experiments. We added this information to the corresponding method section (line 1059).

- I think that more detailed/clearer cartoon(s) of the “geometry” of the experiments (showing the surface+proteins) would help to make the manuscript descriptions far more readable without forcing someone to look for other resources. The *M_w*, dimensions and pI of the proteins would be useful information too – they are vital for supporting/explaining some of the results.

We now include a detailed illustration of our experiment in Figure S1a. Additionally we added the protein parameters to Supplementary Table 2.

Protein	M_w [Da]	Width	Length [nm]	pI
FtsA WT	45329.97	~5nm	~7nm	5.84
FtsA R286W	45359.99	~5nm	~7nm	5.75
FtsZ	40192.72	~4nm	~6nm	4.63

Figure 2: Illustration of the experimental including the dimensions of the proteins used in our experiments.

In conclusion, this study contains a lot of new work in an area where it is much needed; I read the manuscript with great interest. The authors tried to build their case slowly putting into it a lot of experimental evidence. The fluorescence technique was indeed greatly exploited but the QCM was not; much information is left untouched. Nevertheless, many findings are very interesting and entirely novel and they would make a valuable contribution to the field. In my opinion, the manuscript can be accepted for publication after consideration is given to the above-mentioned Comments.

We thank the reviewer for these kind words and the helpful comments!

Reviewer #2 (Remarks to the Author):

This paper investigates the interaction of FtsA and its variant R268W with lipids, FtsZ and FtsN on a lipid surface. The variant is known to bypass other division proteins and to cause premature division and to favor the monomeric form of FtsA due to its reduced ability to polymerize. FtsA binds to membranes in the absence of FtsZ and assembles with FtsZ in its presence. R268W migrates together with FtsZ at a range of concentrations whereas the wt distributes on the membranes at higher concentrations. The affinity of WT and variant for membranes is similar as was already shown by Krupka et al., 2017. Affinities for the C-terminal peptide of FtsZ seemed to be similar for WT and variant. The hypothesis is that WT FtsA forms minirings on the lipid layer that limits the number of associated FtsZ filaments, whereas R268W assembles in aligned structures that allow more FtsZ filaments to bind at the higher FtsA proteins concentrations (based on EM images from Krupka et al 2017). R268W recruited the cytoplasmic domain of FtsN faster than wt FtsA without having a higher affinity for FtsN. The new information of this paper is that R268W has a much shorter residence time and diffuses much faster on the lipid bilayers than the WT and that WT and variant have the same affinity for the C-term of FtsZ and for the cytoplasmic domain of FtsN. Based on the EM images of Krupka, the N-term of FtsA should be inside the FtsA ringlets and CY3 and Cy5 could be very close together (3.5 nm) from which a very high FRET efficiency is to be expected. However, this is diluted since maximally 50% of the molecules contribute to the FRET signal.

We agree that the expected FRET efficiency could be higher due to miniring formation. However, with our labelling approach we only achieve a labeling efficiency of ~65-70%, which decreases the possible maximum FRET efficiency. In the case of FtsA wt, we reach nearly 50% FRET efficiency at concentrations higher than 1.5 μ M (shown in Fig. S3k), which is roughly the maximal value to we can expect under these conditions. The lower FRET efficiency observed for FtsA R286W is the result of its lower degree of oligomerization and as a consequence its faster turnover on the membrane.

At high FtsA concentrations molecules from different ringlets can also contribute to the FRET explaining the increase in FRET efficiency as is observed. The R268W mutant has a much lower FRET signal, which might be explained by its short residence time on the membranes. When the protein is permanently attached to the membrane by a His-tagged lipid it still has a very low FRET efficiency (about 10%). Based on the hypothesis that it forms arcs and bundles and collects FtsZ that also forms bundles and the drawing in the model that predicts that every R268W protein is bound to a FtsZ molecule, I would expect the FRET efficiency to be higher. I have the impression that R268W is not associating with itself and that FtsZ has no effect on the association. In that sense it is a bit strange to depict in the model a row of R268W molecules that clearly interact.

We have modified the illustration and hope that the reviewer finds it improved.

We want to point out that we compared the FRET efficiency of FtsA and FtsA R268W with fluorescently labeled His-SUMO on membranes with defined amounts of Tris-NTA lipids (Fig. 4g). As this protein cannot self-interact, we used it as a negative control in our experiments. Importantly, for His-SUMO, we cannot observe any FRET, nor a reduction in lateral mobility, even at the maximum density of the surface-attached protein. This shows that FtsA R268W can form oligomers, although to a lesser degree than the wildtype protein.

However, we clearly see enhanced FtsA R286W self-interaction in the presence of FtsZ and FtsN_{cyto} which nourishes our model for better aligned FtsA R286W structures beneath FtsZ. Compared to the proteins with native membrane binding, the His-tagged variants show lower FRET, suggesting that the amphipathic helix can somehow contribute to FtsA oligomerization, possibly by allowing for a conformational change after binding to FtsZ. Importantly, we now also include experiments with the His-tagged FtsAs in the presence of FtsN_{cyto} (see our reply below).

As soon as the cytoplasmic domain of FtsN is added WT and R268W become very similar with respect to FRET efficiency and diffusion rate. Only the off rate of R286W is still higher if it is not forced to bind to the membrane. Binding of FtsN apparently reorientates the R268W molecules and presumably also the WT molecules. As mentioned in the discussion, it has recently been found that FtsN stimulates the conversion of FtsA ringlets to double filaments.

It is a pity that the authors did not add FtsN to the membrane immobilized system as well.

We agree with the reviewer that this is an exciting experiment. Following their suggestion, we performed experiments with His-tagged FtsAs in the presence of FtsN_{cyto}-His on supported membrane containing 1.5% Tris-NTA lipids. Our results further support our findings that FtsN does not depolymerize FtsA oligomers and even enhances self-interaction in the case of FtsA R286W (see Figure 3 in this document as well as Fig. S5 in our manuscript). Specifically, we find:

- 1. The presence of FtsN_{cyto}-His rendered the His-tagged FtsAs immobile. The measured diffusion coefficients were close to zero and we could not observe complete recovery of the fluorescence signal after photobleaching (Fig. S5c, d and Fig S5 g, h).*
- 2. When we added FtsZ, FtsN_{cyto}-His and His-tagged FtsA formed a filamentous pattern on the membrane. However, this pattern was very static and did not show the kind of dynamic behavior as seen for native FtsAs or His-FtsA without FtsN_{cyto}-His (Fig S5a, b), most likely because any exchange of the membrane anchor, either via diffusion or membrane exchange is abolished. FtsZ was still turning over with a mean recovery time of around 9.5 s.*
- 3. FRET of FtsA R286W-His increased more than twofold to similar levels as seen for FtsA WT-His. As for the native FtsA WT, the FRET signal was unchanged for the His-tagged proteins in the presence of FtsN_{cyto} (Fig S5e, f), indicating that FtsA stays polymeric.*

Although these experiments further support our finding that FtsN enhances FtsA self-interaction, it is important to point out that they have some limitations: As both, FtsN_{cyto}-His and FtsA-His bind the membrane via His6-tags, they compete for the interaction with Ni²⁺-chelating lipids. In the experiments shown, SLBs with 1.5% Tris-NTA lipids were exposed to equimolar amounts of FtsN_{cyto}-His and FtsA-His in the buffer, however we cannot accurately control the corresponding densities on the membrane surface.

However, these experiments shed more light on the biochemical properties of FtsA oligomerization and qualitatively support our finding that FtsN_{cyto}-His enhances self-interaction of FtsA on membranes. We therefore decided to include these new data in the supplemental information of manuscript (Fig. S5).

Figure 3: New experiments on the effect of membrane-bound FtsN_{cyto}His on FtsA-His are now shown in Figure S5.

Overall, I think that the interpretation of the data is to a large extent correct and fit the model (maybe some adjustments can be made bases on the thoughts above and below).

Minor remarks:

In Fig. 1c at high FtsA concentrations one can see that the fluorescence is completely distributed over the membrane apart from some bright spots. This suggests that FtsA cannot oligomerize under these conditions. Perhaps it is good to mention explicitly that the minirings of FtsA are less than 20 nm in diameter and therefore not visible on the lipid bilayer?

*We want to apologize that our manuscript was not clear in this point. The reviewer is correct that we cannot resolve FtsA minirings with a diameter of only 20 nm by TIRF microscopy. Accordingly, a homogeneous fluorescence on the membrane surface does **not** suggest the absence of oligomers. We now modified the text accordingly (lines 171-175).*

Also, the FtsZ filaments look different in the image at higher FtsA concentration, fewer ringlets. In figure S1 you show that the dynamics of FtsZ do not change. Does this mean that the difference in morphology is just coincidentally? In Fig 2b I see the same, so no coincidence?

The reviewer is correct. As observed in this work and before (Loose & Mitchison NCB 2014), high concentrations of FtsA can negatively affect the organization of FtsZ filaments into bundles. At FtsA WT concentrations above 0.4 μ M we can indeed observe that the FtsZ pattern is slightly disturbed. However, we can still observe bundles of treadmilling FtsZ filament, whereas FtsA WT is homogeneously distributed along the membrane. We have modified the manuscript and hope the reviewer finds it improved (lines 112-116).

Figure S1i Y-ax should be autocorrelation.

Corrected the typo.

Line 142: maybe better? In particular, previous yeast two-hybrid experiments suggested that FtsA R286W has a higher affinity for FtsZ than the wild-type protein.

Rephrased sentence.

Line 150: The difference in residence time of the FtsZ peptide on FtsA WT or mutant is not statistically significant. This is caused by the large SD of the WT. If the WT would have the same SD as the mutant, the difference in dwelling time would be about 18%. Would that be sufficient to explain a higher affinity or is it something that can be ignored? FtsZ would never be a monomer when it makes a functional interaction with FtsA. As a polymer the FtsZ concentration would be locally very high, then a small increase in affinity might be meaningful?

A higher affinity of the FtsZ CTP towards FtsA WT would result in an enhanced recruitment of FtsZ filaments to the membrane. However, we observe the opposite: FtsA R286W allows for a higher intensity of FtsZ filaments (Fig. S1f), most likely due to an increased packing density on the membrane.

The reviewer is correct that during the recruitment of FtsZ filaments avidity effects will play an important role. We decided to only test the binding of the C-terminal peptide to avoid any effect of FtsZ polymerization and to specifically test the interaction between a single FtsZ monomer and FtsA.

As a complementary method, we also tried measuring the affinity between FtsA and FtsZ CTP using Microscale Thermophoresis. However, as the affinity appears to be very weak we did not succeed with these experiments. We also measured the lifetimes of FtsZ monomers within the filaments by tracking single FtsZ molecules and did not find a significant difference when either FtsA WT or R286W was used (Fig. S1e).

Fig 2c and d. The X-as title of c is overlapped a bit by d.

We fixed this.

Line 249: the number of digits indicates precision; therefore, it seems logical to write that FtsA retention time at 0.8 μ M is 9.4 ± 0.3 s. Otherwise, you suggest that the precision is better at higher FtsA concentrations for which I do not immediately see a reason...

Corrected.

Line 248: Unfortunately, I do not have access to your paper “Single-molecule measurements to study polymerization dynamics of FtsZ-FtsA copolymers. “ which should describe how you measure residence time. Consequently, I do not immediately understand that you measure absence of diffusion for FtsA WT $0.004 \text{ um}^2/\text{s}$ but a residence time of 10 sec. based on your movie I would define residence time. The time FtsA is bound to the lipid layer before it dissociates into the medium. If I am right, it might be helpful for the reader to add such a definition to the paper?

This reference describes an approach to estimate the contribution of fluorophore bleaching to the observed mean residence time. Here, we record single molecule movies of 1-2 minute lengths with different acquisition rates (0.125/0.25/0.5/1/2 sec) and therefore different contributions of bleaching to the observed single molecule lifetimes. The actual residence of the protein can then be estimated by plotting the measured residence time against the acquisition and fitting a linear function to this relationship. A similar approach has been previously described by Gebhardt et al. Nature Methods 2013, which we now also provide as a reference

Line 295: Förster. Keep pressing your finger on the o key and a menu will pop up that allows you to choose different o's.

Changed o to ö.

Line 260: either Cy-5 (acceptor) or Cy3 (donor). Otherwise, the reader must search for this information, which makes it difficult to understand fig 3 d and so on.

Added acceptor and donor

Fig. S3C, the legend and word FRET-threshold are for some reason not completely visible.

Thanks a lot, we made sure that all information is visible in the updated version.

In previous experiments it was shown that the FtsA variants have the same affinity for the membrane. In the cuvette experiment where you add suvs and FtsA variants and assuming that the labeling efficiency of both proteins was the same, you would expect the same amount of FRET if these proteins would be able to interact with each other in the same manner, is it not? According to the literature FtsA forms minirings and R286W forms tightly packed filaments and arcs. Yet the WT achieves a higher FRET efficiency. Initially I found that counterintuitive. But since you measure the time average FRET, the lower retention time of R268W probably explains the lower FRET despite the higher change of Cy3 and 5 to see each other in packed filaments than in ringlets.

We agree with the reviewer that the shorter residence time counterbalances a tighter packing of monomers on the membrane surface. In addition, the FtsA R286W assemblies are likely less ordered than the well-structured miniring arrays of FtsA WT. We added this interpretation of our observation to the Figure caption of Fig. S3.

in the manuscript, you assume that FtsA behaves as described by Krupka in 2017 using EM. But you have the proteins labeled. How do you know that this is not affecting the assembled conformation of the proteins?

The functionality of labeled proteins is an important point that always needs to be considered in fluorescence microscopy experiments. In our experiments, we make sure that labeled proteins are active using the following measures:

1. We only use small organic dyes instead of fluorescent fusion proteins
2. We choose fluorophores that were found to have minimal interactions with membranes (Hughes et al PLoS One 2013).
3. The fluorophore is attached to a position of the protein that does not take part in interaction with other proteins or the membrane.
4. We make sure that different fluorophores at the protein of interest lead to the same behavior of the protein, i.e. that our observation are not dye-specific.

In addition, in experiments with all three components (FtsZ, FtsA and FtsZ), there was always one that was not labeled, i.e. if we assume the protein to be either **dark**, labeled with a **red** or **green** dye:

Exp. 1	FtsA	FtsZ	FtsN
Exp. 2	FtsA	FtsZ	FtsN
Exp. 3	FtsA	FtsZ	FtsN

If the fluorophore on FtsA, had a big impact on the observed behavior, we would expect, that the spatiotemporal pattern of labeled FtsN looks different in Experiments 2 and 3 or that FtsZ would show different behavior in Experiments 1 and 2. Importantly, this was not the case for either of the two versions of FtsA.

The difference in FRET suggest that these variants interact differently (as expected). Have you thought of trying alphafold (- colab.research.google.com/githu...Xdjf18) with the variants to see whether any structural differences might be able to explain your results?

Thanks for the suggestion. We now performed the suggested comparison of predicted structures of FtsA WT and FtsA R286W modelled by Alphafold.

Figure 4: Overlay of the predicted structures of FtsA WT (green) and FtsA R286W (cyan). The red/magenta depicts amino acid 286. The structures were modelled using AlphaFold2, superimposed with Chimera and compared with MatchMaker. The RMSD was determined to be 0.378 (Q-Score of 0.979).

From the predicted structures however, we were not able to discern a significant difference between both proteins. The RMSD was low and the differences within the margin of error.

Interestingly, the FRET signal is almost the same for WT and R286W in the presence of FtsZ and FtsN as if binding of FtsN forces the two proteins in a similar orientation/interaction. Also, the diffusion coefficients are more similar. You could try to look at the combination of the cytoplasmic domain of FtsN and the FtsA variants in alpha-fold as well.

Figure 5: Overlay of the predicted structures of FtsA WT (green) and FtsA WT + FtsN (FtsA in cyan, FtsN in magenta). The 30xGly linker to connect FtsN to FtsA's C-terminal end is coloured white. The structures were modelled using AlphaFold2, superimposed with Chimera and compared with MatchMaker. The RMSD was determined to be 0.415 (Q-Score of 0.826).

From the predicted structures for FtsA WT and FtsA WT + FtsN, we were not able to discern a significant change in the structure of FtsA WT. The Q-Score is lower, but this is due to the presence of the linker at the C-terminal end, which leads to stronger deviation of the FtsA structures at these positions. In general, the C-terminal Helix of FtsA cannot be properly modelled using AlphaFold and appears like a cork-screw pointing outwards of the structure.

However, we tried to model dimers of FtsA WT and FtsA R286W, which we performed with AlphaFold2 Advance. Modelling of higher order oligomers (i.e. dodecamers like the rings observed for FtsA WT) unfortunately is not possible with the current AlphaFold version, as the length limit is 1400 amino acids.

Figure 6: Overlay of FtsA WT (green) and FtsA R286W dimers (cyan). Amino acid 286 is labelled in magenta. The structures were modelled using AlphaFold2, superimposed with Chimera and compared with MatchMaker. The RMSD was determined to be 0.181 (Q-Score of 0.956).

In the dimer prediction, the models deviate even less from each other. The Arginine or the Tryptophan in position 286 align perfectly with each other and from these predictions it is very hard to find differences stemming from single amino acid substitutions. AlphaFold works with already deposited structures and forms an average of an expected structure. Thus, small changes are most likely overseen and cannot be identified using this approach.

In the model that explains your observations, you assume that FtsA form ringlets. According to the paper of Krupka these are 20 nm 5 FtsZ molecules span 20 nm, so in the drawing you have every second FtsZ molecule binds an FtsA, but it should be every 5th FtsZ molecule binds an FtsA.

We changed the drawing in the model according to the reviewers' suggestion.

Line 873: FtsZ was added instead of were added?

We changed "were" to "was".

Reviewer #3 (Remarks to the Author):

The authors present an interesting study of the in vitro assembly dynamics of E. coli FtsA, focussing on the mechanism of assembly and co-treadmilling with FtsZ. In particular they investigate how these assembly dynamics are modulated by the FtsA R286W gain of function mutant, which can support division without E. coli's other membrane anchor, ZipA, and by interactions with the division-activating protein FtsN.

The authors principal findings are separated into those concerning the function of the FtsA R286W mutant, and those concerning FtsN activation of FtsA.

R286W

They find that FtsA R286W colocalizes and co-treadmills much more precisely with FtsZ over a wider range of FtsA concentrations. Fitting with this they find that FtsA R286W also turns over much quicker from the membrane, and diffuses much faster. Altogether this supports a model where inhibition of more stable large arcs/ circles of FtsA common in the WT is inhibited in this mutant, and this allows FtsA R286W to track FtsZ localization more accurately, likely explaining why it acts as a gain of function mutant. A preprint from the Lowe lab supports the underlying structural hypothesis; this study provides excellent complementary analysis linking the new structural information to the resulting protein localization dynamics which are known to be central to the function of both FtsA and FtsZ.

FtsN

The FtsN results I found a little trickier to follow. Partly, I think this is because the implications of the results are a bit more complex. Interestingly the authors find that FtsN probably does lead to disassembly of FtsA oligomers, as previously speculated – rather, self interaction, measured via FRET, actually increases. However, the increased self interaction actually leads to faster FtsA turnover and diffusion on the membrane (Figure 3), which naively is probably the opposite of what one would predict - larger assemblies should be more stable – but again fits well with the recent Lowe lab work, which shows addition of FtsN to FtsA forms assembly of actin like double filaments instead of mini-rings. Excitingly, the Radler work suggests then that the FtsA+FtsN double filaments are likely much more dynamic than the FtsA arcs, allowing more accurate co-treadmilling/ co-assembly with FtsZ at the septum – as summarized in their cartoon.

We are sorry that this part of the manuscript was not 100% clear. We find that FtsN increases self-interaction of FtsA, slightly increases its residence time, i.e. decreases the off-rate and slows down diffusion. To avoid any confusion, we added new panels to Figure S3f-h to specifically show the influence of each component of the protein system on these parameters.

Overall I really enjoyed reading this work and I recommend accepting it once the below minor criticisms are addressed. I have essentially no technical criticisms of the manuscript of substantial consequence – I was impressed with the level of experimental care and detail, and the overall clarity of the figures.

We thank the reviewer for the kind words!

Comments to be addressed:

Please make clear that the work has been done in *E. coli*. In the title – please change to “In vitro reconstitution of *Escherichia coli* divisome activation” or similar. Also clarify in the abstract. In the introduction, eg division is introduced as “Divisome assembly is initiated by the simultaneous accumulation of FtsZ, FtsA and ZipA at midcell, where they organize into the Z-ring, a composite cytoskeletal structure of treadmilling filaments at the inner face of the cytoplasmic membrane (Fig. 1a).” with no mention of the organism. In fact the organism is only mentioned once rather late (line 56) in the intro. The *ftsA* R286W gain of function mutant has, to the best of my knowledge, only been characterized in *E. coli*, FtsN is limited to the Proteobacteria, and ZipA is primarily characterized only in the context of *E. coli*. Therefore it is not justified to draw inference beyond this organism - although I would certainly be interested to hear the authors speculation on implications for organisms which lack FtsN. This over-generalism is quite annoying to the non-*E.-coli* biologist! Please address this.

We agree with the reviewer and have changed the title and manuscript according to their suggestion. We also agree that a systematic comparison of the mechanisms of divisome activation of different organisms would be very valuable to identify general themes and unique mechanisms to control cell division. In fact, Gram-positive bacteria lack FtsN and FtsA seems to be dispensable suggesting a completely different mechanism of divisome activation. To our knowledge, it is at the moment not clear how these cells control the activity of the divisome. We now address this lack of knowledge in the concluding sentence of our discussion.

Implications of the study for activation of FtsA. I struggled a bit with understanding how the FtsA R286W findings relate to the FtsA WT + FtsN experiments and specifically to the mechanisms of FtsA activation. The authors mention early “By comparing the properties of wildtype FtsA and FtsA R286W, a hyperactive mutant that represents the activated state of the protein” (line 75) – I don’t think this is accurate, and also I think their results argue against the simplest version of this statement in some quite interesting ways. FtsA R286W has been characterized as a gain of function mutant which can support division in the absence of ZipA. It has previously been speculated that this may be because FtsA R286W is the activated form of FtsA. Actually the recent Lowe preprint seems to argue against this.

If the active form of FtsA is double filaments, and the inactive form is arcs, FtsA R286W without FtsN definitely does not look like the active form, instead it looks like more compact arcs (Figure 2), which transition to the active form more easily on addition of FtsN.

Furthermore the authors results also seem to argue against this simplest interpretation. FtsA R286W both turns over on the membrane and diffuses incredibly fast compared to WT FtsA (Fig 2b). If FtsA R286W is the active state of FtsA, and FtsN indeed promotes activation of FtsA, then one would expect the dynamics of FtsN to FtsA WT to approach the fast turnover of FtsA R286W – actually addition of FtsN to FtsA appears to cause the turnover and diffusion of FtsA to slow down a little bit (Fig 2i,j). This seems to be the opposite to the “FtsA R286W is just FtsA’s active state” model.

Certainly the authors should clarify what they mean on line 75, but I would also suggest that the authors consider adding some discussion of the implications of their study for the relationship between FtsA R286W and the FtsA WT active state. Also to discuss the lack of similarity between the FtsA + FtsN and FtsAR286W results – it may be that their interpretation of these data are very different to mine but I would like to know how they do interpret these very interesting results

We fully agree with the reviewer. The starting point of this study were previous interpretations of in vivo experiments that suggested that FtsA R286W would behave identical as the wildtype proteins in complex with FtsN. Our observations as well as that of Jan Löwe's group now clearly show that this simple interpretation is incorrect. Instead, it has now become clear FtsA as well FtsA R286W form similar double-filaments with FtsN with similar FRET efficiencies, while in the absence of FtsN they assemble into different polymeric structures, i.e. rings or arcs. We have updated the manuscript accordingly and I hope the reviewer finds it improved.

Related point: I found it hard to make key comparisons of results with FtsA to results with FtsA R286W in the left and right panels of Figure 3hij. I'm not sure what the best solution is but maybe adding gridlines to the graphs would help compare between them? Are there any summary statistics that can be extracted from those concentration dependent data to aid comparison between the 8 different conditions. Please consider using more distinct colour maps in Fig 3hij, the different shades of blue/ purple, and different shades of green or red I struggle to tell the difference between, which made it quite tricky to interpret those graphs.

We tried to make the colors more distinct and added some grid-lines which should simplify the comparisons at 30% FRET Efficiency, D_{coeff} of $0.01 \mu\text{m}^2/\text{s}$ and an off binding rate of 0.1 s^{-1} . Additionally, we added new panels in Fig. S3 (Fig. S3f-S3h) where we show boxplots of two different concentrations of FtsA WT/R286W. These boxplots show the effect of FtsN, FtsZ and FtsZ/FtsN on the FRET/ D_{coeff} /Off rate and we hope they simplify the comparison.

REVIEWERS' COMMENTS

Reviewer #1 (Remarks to the Author):

Accept as is

Reviewer #2 (Remarks to the Author):

I have read the replies of the authors to the comments of the reviewers and think that all issues were clarified. I am also happy that the authors managed to include FtsN to the membrane immobilized system. The manuscript was already very interesting, the experiments very complete and well executed. The data interpretation was correct. Some aspects were not entirely clear to the reader in the first submission and those issue have been improved. I have no further comments and hope that the paper will be published.

Reviewer #3 (Remarks to the Author):

The authors have fully addressed all of my comments and I support publication of the manuscript.